# Ambient Diffusion:
# Learning Clean Distributions from Corrupted Data

Giannis Daras
UT Austin
giannisdaras@utexas.edu

Kulin Shah
UT Austin
kulinshah@utexas.edu

Yuval Dagan
UC Berkeley
yuvald@berkeley.edu

Aravind Gollakota
Apple
aravindg@cs.utexas.edu

Alexandros G. Dimakis
UT Austin
dimakis@austin.utexas.edu

Adam Klivans
UT Austin
klivans@utexas.edu

## Abstract

We present the first diffusion-based framework that can learn an unknown distribution using only highly-corrupted samples. This problem arises in scientific applications where access to uncorrupted samples is impossible or expensive to acquire. Another benefit of our approach is the ability to train generative models that are less likely to memorize any individual training sample, since they never observe clean training data. Our main idea is to introduce *additional measurement distortion* during the diffusion process and require the model to predict the original corrupted image from the further corrupted image. We prove that our method leads to models that learn the conditional expectation of the full uncorrupted image given this additional measurement corruption. This holds for any corruption process that satisfies some technical conditions (and in particular includes inpainting and compressed sensing). We train models on standard benchmarks (CelebA, CIFAR-10 and AFHQ) and show that we can learn the distribution even when all the training samples have $90\%$ of their pixels missing. We also show that we can finetune foundation models on small corrupted datasets (e.g. MRI scans with block corruptions) and learn the clean distribution without memorizing the training set.

## 1 Introduction

Diffusion generative models [54, 23, 57] are emerging as versatile and powerful frameworks for learning high-dimensional distributions and solving inverse problems [34, 11, 36, 28]. Numerous recent developments [58, 30] have led to text conditional foundation models like Dalle-2 [45], Latent Diffusion [50] and Imagen [52] with an incredible performance in general image domains. Training these models requires access to high-quality datasets which may be expensive or impossible to obtain. For example, direct images of black holes cannot be observed [12, 19] and high quality MRI images require long scanning times, causing patient discomfort and motion artifacts [28].

Recently, Carlini et al. [8], Somepalli et al. [55], and Jagielski et al. [27] showed that diffusion models can memorize examples from their training set. Further, an adversary can extract dataset samples given only query access to the model, leading to privacy, security and copyright concerns. For many applications, we may want to learn the distribution but not individual training images e.g. we might want to learn the distribution of X-ray scans but not memorize images of specific patient scans from the dataset. Hence, we may want to introduce corruption as a design choice. We show that it is possible to train diffusions that learn a distribution of clean data by only observing highly corrupted samples.

37th Conference on Neural Information Processing Systems (NeurIPS 2023).

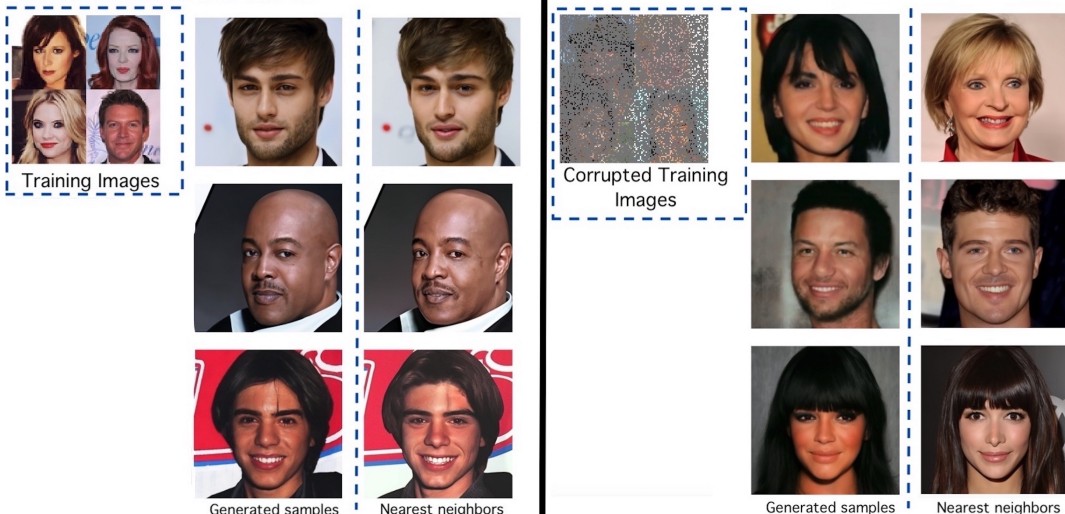

Training Images

Generated samples · Nearest neighbors

Corrupted Training Images

Generated samples · Nearest neighbors

Figure 1: **Left panel:** Baseline method of vanilla finetuning Deepfloyd IF using 3000 images from CelebA-HQ. We show generated sample images and nearest neighbors from the finetuning set. As shown, the generated samples are often near-identical copies from training data. This verifies related work Carlini et al. [8], Somepalli et al. [55], and Jagielski et al. [27] that pointed out that diffusions often generate training samples. **Right panel:** We finetune the same foundation model (Deepfloyd IF) using our method and 3000 highly corrupted training images. The corruption adds noise and removes 80 percent random pixels. We show generated samples and nearest neighbors from the training set. Our method still learns the clean distribution of faces (with some quality deterioration, as shown) but does not memorize training data. We emphasize that our training is performed without ever accessing clean training data.

**Prior work in supervised learning from corrupted data.** The traditional approach to solving such problems involves training a restoration model using supervised learning to predict the clean image based on the measurements [47, 48, 63, 41]. The seminal Noise2Noise [39] work introduced a practical algorithm for learning how to denoise in the absence of any non-noisy images. This framework and its generalizations [5, 38, 59] have found applications in electron microscopy [16], tomographic image reconstruction [62], fluorescence image reconstruction [65], blind inverse problems [20, 5], monocular depth estimation and proteomics [6]. Another related line of work uses Stein's Unbiased Risk Estimate (SURE) to optimize an unbiased estimator of the denoising objective without access to non-noisy data [18]. We stress that the aforementioned research works study the problem of *restoration*, whereas are interested in the problem of *sampling* from the clean distribution. Restoration algorithms based on supervised learning are only effective when the corruption level is relatively low [15]. However, it might be either not possible or not desirable to reconstruct individual samples. Instead, the desired goal may be to learn to *generate* fresh and completely unseen samples from the distribution of the uncorrupted data but *without reconstructing individual training samples*.

Indeed, for certain corruption processes, it is theoretically possible to perfectly learn a distribution only from highly corrupted samples (such as just random one-dimensional projections), even though individual sample denoising is usually impossible in such settings.

AmbientGAN [7] showed that general $d$ dimensional distributions can be learned from *scalar* observations, by observing only projections on one-dimensional random Gaussian vectors, in the infinite training data limit. The theory requires an infinitely powerful discriminator and hence does not apply to diffusion models. MCFlow [49] is a framework, based on a variant of the EM algorithm, that can be used to train normalizing flow models from missing data. Finally, MIWAE [42] and Not-MIWAE [26] are frameworks to learn deep latent models (e.g. VAEs) from missing data when the corruption process is known or unknown respectively.

**Our contributions.** We present the first diffusion-based framework to learn an unknown distribution $\mathcal{D}$ when the training set only contains highly-corrupted examples drawn from $\mathcal{D}$. Specifically, we consider the problem of learning to sample from the target distribution $p_0(\boldsymbol{x}_0)$ given corrupted

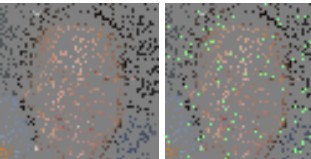 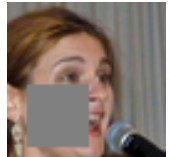 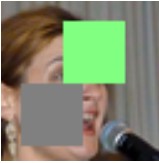

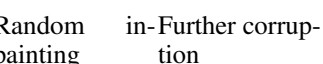

| Random in-
painting | Further corrup-
tion | Block inpaint-
ing | Further corrup-
tion |

Figure 2: Illustration of our method: Given training data with deleted pixels, we corrupt further by erasing more (illustrated with green color). We feed the learner the further corrupted images and we evaluate it on the originally observed pixels. We can do this during training since the green pixel values are known to us. The score network learner has no way of knowing whether a pixel was missing from the beginning or whether it was corrupted by us. Hence, the score network learns to predict the clean image everywhere. Our method is analogous to grading a random subset of the questions in a test, but the students not knowing which questions will be graded.

samples $A\boldsymbol{x}_0$ where $A \sim p(A)$ is a random corruption matrix (with known realizations and prior distribution) and $\boldsymbol{x}_0 \sim p_0(\boldsymbol{x}_0)$. Our main idea is to introduce *additional measurement distortion* during the diffusion process and require the model to predict the original corrupted image from the further corrupted image.

- We provide an algorithm that provably learns $\mathbb{E}[\boldsymbol{x}_0|\tilde{A}(\boldsymbol{x}_0 + \sigma_t\boldsymbol{\eta}), \tilde{A}]$, for all noise levels $t$ and for $\tilde{A} \sim p(\tilde{A} \mid A)$ being a further corrupted version of $A$. The result holds for a general family of corruption processes $A \sim p(A)$. For various corruption processes, we show that the further degradation introduced by $\tilde{A}$ can be very small.
- We use our algorithm to train diffusion models on standard benchmarks (CelebA, CIFAR-10 and AFHQ) with training data at different levels of corruption.
- Given the learned conditional expectations we provide an approximate sampler for the target distribution $p_0(\boldsymbol{x}_0)$.
- We show that for up to 90% missing pixels, we can learn reasonably well the distribution of uncorrupted images. We outperform the previous state-of-the-art AmbientGAN [7] and natural baselines.
- We show that our models perform on par or even outperform state-of-the-art diffusion models for solving certain inverse problems even without ever seeing a clean image during training. Our models do so with a single prediction step while our baselines require hundreds of diffusion steps.
- We use our algorithm to finetune foundational pretrained diffusion models. Our finetuning can be done in a few hours on a single GPU and we can use it to learn distributions with a few corrupted samples.
- We show that models trained on sufficiently corrupted data do not memorize their training set. We measure the tradeoff between the amount of corruption (that controls the degree of memorization), the amount of training data and the quality of the learned generator.
- We open-source our code and models: https://github.com/giannisdaras/ambient-diffusion.

## 2 Background

Training a diffusion model involves two steps. First, we design a corruption process that transforms the data distribution gradually into a distribution that we can sample from [58, 14]. Typically, this corruption process is described by an Ito SDE of the form: $\mathrm{d}\boldsymbol{x} = \boldsymbol{f}(\boldsymbol{x}, t)\mathrm{d}t + g(t)\mathrm{d}\boldsymbol{w}$, where $\boldsymbol{w}$ is the standard Wiener process. Such corruption processes are *reversible* and the reverse process is also described by an Ito SDE [3]: $\mathrm{d}\boldsymbol{x} = \left(\boldsymbol{f}(\boldsymbol{x}, t) - g^2(t)\nabla_{\boldsymbol{x}} \log p_t(\boldsymbol{x})\right)\mathrm{d}t + g(t)\mathrm{d}\boldsymbol{w}$. The designer of the diffusion model is usually free to choose the drift function $\boldsymbol{f}(\cdot, \cdot)$ and the diffusion function $g(\cdot)$. Typical choices are setting $\boldsymbol{f}(\boldsymbol{x}, t) = \boldsymbol{0}, g(t) = \sqrt{\frac{\mathrm{d}\sigma_t^2}{\mathrm{d}t}}$ (Variance Exploding SDE) or setting $\boldsymbol{f}(x, t) = -\beta(t)\boldsymbol{x}, g(t) = \sqrt{\beta(t)}$ (Variance Preserving SDE). Both of these choices lead to a Gaussian terminal distribution and are equivalent to a linear transformation in the input. The goal of diffusion model training is to learn the function $\nabla_{\boldsymbol{x}} \log p_t(\boldsymbol{x})$, which is known as the score function.

To simplify the presentation of the paper, we will focus on the Variance Exploding SDE that leads to conditional distributions $\boldsymbol{x}_t = \boldsymbol{x}_0 + \sigma_t \boldsymbol{\eta}$.

Vincent [61] showed that we can learn the score function at level $t$ by optimizing for the score-matching objective:

$$J(\theta) = \frac{1}{2}\mathbb{E}_{(\boldsymbol{x}_0, \boldsymbol{x}_t)} ||\boldsymbol{h}_\theta(\boldsymbol{x}_t, t) - \boldsymbol{x}_0||^2. \tag{2.1}$$

Specifically, the score function can be written in terms of the minimizer of this objective as:

$$\nabla_{\boldsymbol{x}_t} \log p_t(\boldsymbol{x}_t) = \frac{\boldsymbol{h}_{\theta^*}(\boldsymbol{x}_t, t) - \boldsymbol{x}_t}{\sigma_t}. \tag{2.2}$$

This result reveals a fundamental connection between the score-function and the best restoration model of $\boldsymbol{x}_0$ given $\boldsymbol{x}_t$, known as Tweedie's Formula [17]. Specifically, the optimal $\boldsymbol{h}_{\theta^*}(\boldsymbol{x}_t, t)$ is given by $\mathbb{E}[\boldsymbol{x}_0|\boldsymbol{x}_t]$, which means that

$$\nabla_{\boldsymbol{x}_t} \log p_t(\boldsymbol{x}_t) = \frac{\overbrace{\mathbb{E}[\boldsymbol{x}_0|\boldsymbol{x}_t]}^{\text{best restoration}} - \boldsymbol{x}_t}{\sigma_t}. \tag{2.3}$$

Inspired by this restoration interpretation of diffusion models, the Soft/Cold Diffusion works [14, 4] generalized diffusion models to look at non-Markovian corruption processes: $\boldsymbol{x}_t = C_t \boldsymbol{x}_0 + \sigma_t \boldsymbol{\eta}$. Specifically, Soft Diffusion proposes the Soft Score Matching objective:

$$J_{\text{soft}}(\theta) = \frac{1}{2}\mathbb{E}_{(\boldsymbol{x}_0, \boldsymbol{x}_t)} ||C_t \left(\boldsymbol{h}_\theta(\boldsymbol{x}_t, t) - \boldsymbol{x}_0\right)||^2, \tag{2.4}$$

and shows that it is sufficient to recover the score function via a generalized Tweedie's Formula:

$$\nabla_{\boldsymbol{x}_t} \log p_t(\boldsymbol{x}_t) = \frac{C_t \mathbb{E}[\boldsymbol{x}_0|\boldsymbol{x}_t] - \boldsymbol{x}_t}{\sigma_t}. \tag{2.5}$$

For these generalized models, the matrix $C_t$ is a design choice (similar to how we could choose the functions $\boldsymbol{f}, g$). Most importantly, for $t = 0$, the matrix $C_t$ becomes the identity matrix and the noise $\sigma_t$ becomes zero, i.e. we observe samples from the true distribution.

## 3 Method

As explained in the introduction, in many cases we do not observe uncorrupted images $\boldsymbol{x}_0$, either by design (to avoid memorization and leaking of sensitive data) or because it is impossible to obtain clean data. Here we study the case where a learner only has access to linear measurements of the clean data, i.e. $\boldsymbol{y}_0 = A\boldsymbol{x}_0$, and the corruption matrices $A : \mathbb{R}^{m \times n}$. We note that we are interested in non-invertible corruption matrices. We ask two questions:

1. Is it possible to learn $\mathbb{E}[\boldsymbol{x}_0|A(\boldsymbol{x}_0 + \sigma_t \boldsymbol{\eta}), A]$ for all noise levels $t$, given only access to corrupted samples $(\boldsymbol{y}_0 = A\boldsymbol{x}_0, A)$?

2. If so, is it possible to use this restoration model $\mathbb{E}[\boldsymbol{x}_0|A(\boldsymbol{x}_0 + \sigma_t \boldsymbol{\eta}), A]$ to recover $\mathbb{E}[\boldsymbol{x}_0|\boldsymbol{x}_t]$ for any noise level $t$, and thus sample from the true distribution through the score function as given by Tweedie's formula (Eq. 2.3)?

We investigate these questions in the rest of the paper. For the first, the answer is affirmative but only after introducing additional corruptions, as we explain below. For the second, at every time step $t$, we approximate $\mathbb{E}[\boldsymbol{x}_0|\boldsymbol{x}_t]$ directly using $\mathbb{E}[\boldsymbol{x}_0|A\boldsymbol{x}_t, A]$ (for a chosen $A$) and substitute it into Eq. 2.3. Empirically, we observe that the resulting approximate sampler yields good results.

### 3.1 Training

For the sake of clarity, we first consider the case of random inpainting. If the image $\boldsymbol{x}_0$ is viewed as a vector, we can think of the matrix $A$ as a diagonal matrix with ones in the entries that correspond to the preserved pixels and zeros in the erased pixels. We assume that $p(A)$ samples a matrix where each entry in the diagonal is sampled i.i.d. with a probability $1 - p$ to be 1 and $p$ to be zero.

We would like to train a function $\boldsymbol{h}_\theta$ which receives a corruption matrix $A$ and a noisy version of a corrupted image, $\boldsymbol{y}_t = A \underbrace{(\boldsymbol{x}_0 + \sigma_t \boldsymbol{\eta})}_{\boldsymbol{x}_t}$ where $\boldsymbol{\eta} \sim \mathcal{N}(0, I)$, and produces an estimate for the conditional expectation. The simplest idea would be to simply ignore the missing pixels and optimize for:

$$J_{\text{naive}}^{\text{corr}}(\theta) = \frac{1}{2} \mathbb{E}_{(\boldsymbol{x}_0, \boldsymbol{x}_t, A)} \left|\left| A \left(\boldsymbol{h}_\theta(A, A\boldsymbol{x}_t, t) - \boldsymbol{x}_0\right)\right|\right|^2, \tag{3.1}$$

Despite the similarities with Soft Score Matching (Eq 2.4), this objective will not learn the conditional expectation. The reason is that the learner is never penalized for performing arbitrarily poorly in the missing pixels. Formally, any function $\boldsymbol{h}_{\theta'}$ satisfying $A\boldsymbol{h}_{\theta'}(A, \boldsymbol{y}_t, t) = A\mathbb{E}[\boldsymbol{x}_0|A\boldsymbol{x}_t, A]$ is a minimizer.

Instead, we propose to *further corrupt* the samples before feeding them to the model, and ask the model to predict the original corrupted sample from the further corrupted image.

Concretely, we randomly corrupt $A$ to obtain $\tilde{A} = BA$ for some matrix $B$ that is selected randomly given $A$. In our example of missing pixels, $\tilde{A}$ is obtained from $A$ by randomly erasing an additional fraction $\delta$ of the pixels that survive after the corruption $A$. Here, $B$ will be diagonal where each element is 1 with probability $1 - \delta$ and 0 w.p. $\delta$. We will penalize the model on recovering all the pixels that are visible in the sample $A\boldsymbol{x}_0$: this includes both the pixels that survive in $\tilde{A}\boldsymbol{x}_0$ and those that are erased by $\tilde{A}$. The formal training objective is given by minimizing the following loss:

$$J^{\text{corr}}(\theta) = \frac{1}{2} \mathbb{E}_{(\boldsymbol{x}_0, \boldsymbol{x}_t, A, \tilde{A})} \left|\left| A \left(\boldsymbol{h}_\theta(\tilde{A}, \tilde{A}\boldsymbol{x}_t, t) - \boldsymbol{x}_0\right)\right|\right|^2, \tag{3.2}$$

The key idea behind our algorithm is as follows: the learner does not know if a missing pixel is missing because we never had it (and hence do not know the ground truth) or because it was deliberately erased as part of the further corruption (in which case we do know the ground truth). Thus, the best learner cannot be inaccurate in the unobserved pixels because with non-zero probability it might be evaluated on some of them. Notice that the trained model behaves as a denoiser in the observed pixels and as an inpainter in the missing pixels. We also want to emphasize that the probability $\delta$ of further corruption can be arbitrarily small as long as it stays positive.

The idea of further corruption can be generalized from the case of random inpainting to a much broader family of corruption processes. For example, if $A$ is a random Gaussian matrix with $m$ rows, we can form $\tilde{A}$ by deleting one row from $A$ at random. If $A$ is a block inpainting matrix (i.e. a random block of fixed size is missing from all of the training images), we can create $\tilde{A}$ by corrupting further with one more non-overlapping missing block. Examples of our further corruption are shown in Figure 2. In our Theory Section, we prove conditions under which it is possible to recover $\mathbb{E}[\boldsymbol{x}_0|\tilde{A}\boldsymbol{x}_t, \tilde{A}]$ using our algorithm and samples $(\boldsymbol{y}_0 = A\boldsymbol{x}_0, A)$. Our goal is to satisfy this condition while adding minimal further corruption, i.e. while keeping $\tilde{A}$ close to $A$.

### 3.2 Sampling

To sample from $p_0(\boldsymbol{x}_0)$ using the standard diffusion formulation, we need access to $\nabla_{\boldsymbol{x}_t} \log p_t(\boldsymbol{x}_t)$, which is equivalent to having access to $\mathbb{E}[\boldsymbol{x}_0|\boldsymbol{x}_t]$ (see Eq. 2.3). Instead, our model is trained to predict $\mathbb{E}[\boldsymbol{x}_0|\tilde{A}\boldsymbol{x}_t, \tilde{A}]$ for all matrices $A$ in the support of $p(A)$.

We note that for random inpainting, the identity matrix is technically in the support of $p(A)$. However, if the corruption probability $p$ is at least a constant, the probability of seeing the identity matrix is exponentially small in the dimension of $\boldsymbol{x}_t$. Hence, we should not expect our model to give good estimates of $\mathbb{E}[\boldsymbol{x}_0|\tilde{A}\boldsymbol{x}_t, \tilde{A}]$ for corruption matrices $A$ that belong to the tails of the distribution $p(A)$.

The simplest idea is to sample a mask $\tilde{A} \sim p(\tilde{A})$ and approximate $\mathbb{E}[\boldsymbol{x}_0|\boldsymbol{x}_t]$ with $\mathbb{E}[\boldsymbol{x}_0|\tilde{A}\boldsymbol{x}_t, \tilde{A}]$. Under this approximation, the discretized sampling rule becomes:

$$\boldsymbol{x}_{t-\Delta t} = \underbrace{\frac{\sigma_{t-\Delta t}}{\sigma_t}}_{\gamma_t} \boldsymbol{x}_t + \underbrace{\frac{\sigma_t - \sigma_{t-\Delta t}}{\sigma_t}}_{1-\gamma_t} \underbrace{\mathbb{E}[\boldsymbol{x}_0|\tilde{A}\boldsymbol{x}_t, \tilde{A}]}_{\hat{\boldsymbol{x}}_0}. \tag{3.3}$$

This idea works surprisingly well. Unless mentioned otherwise, we use it for all the experiments in the main paper and we show that we can generate samples that are reasonably close to the true distribution (as shown by metrics such as FID and Inception) even with $90\%$ of the pixels missing.

We include pseudocode for our sampling in Algorithm 1. In the Appendix (Section E.4), we ablate an alternative choice for sampling that can lead to slight improvements at the cost of increased function evaluations.

---

**Algorithm 1** Ambient Diffusion Fixed Mask Sampling (for VE SDE)

---

**Input:** noise schedule $\{\sigma_t\}_0^T$, model parameters $\theta$, and step size $\Delta t$ for discretizing the Reverse ODE.

    Sample $\boldsymbol{x}_T \sim \mathcal{N}(0, \sigma_T^2 I)$                                       ▷ Sample initial noise
    Sample $\tilde{A} \sim p(\tilde{A})$                                              ▷ Sample a mask
    $t \leftarrow T$
    **repeat**
        $\boldsymbol{y}_t \leftarrow \tilde{A}\boldsymbol{x}_t$                                    ▷ Create measurements
        $\hat{\boldsymbol{x}}_0 \leftarrow \boldsymbol{h}_\theta(\boldsymbol{y}_t, \tilde{A}, t)$               ▷ Use the network to estimate $\mathbb{E}[\boldsymbol{x}_0 | \tilde{A}\boldsymbol{x}_t, \tilde{A}]$
        $\boldsymbol{x}_{t-\Delta t} \leftarrow \frac{\sigma_{t-\Delta t}}{\sigma_t}\boldsymbol{x}_t + \frac{\sigma_t - \sigma_{t-\Delta t}}{\sigma_t}\hat{\boldsymbol{x}}_0$     ▷ Run one step of the Reverse ODE approximating
    $\mathbb{E}[\boldsymbol{x}_0|\boldsymbol{x}_t]$ with $\hat{\boldsymbol{x}}_0$
        $t \leftarrow t - \Delta t$
    **until** $t \leq \Delta t$
    **return** $\hat{x}_0$

---

We underline that the Fixed Mask Sampler does not come with a theoretical guarantee of sampling from $p_0(\boldsymbol{x}_0)$. In the Appendix, we prove that whenever it is possible to reconstruct $p_0(\boldsymbol{x}_0)$ from corrupted samples, it is also possible to reconstruct it using access to $\mathbb{E}[\boldsymbol{x}_0|A\boldsymbol{x}_t, A]$ (Lemma A.3). However, as stated in the Limitations section, we were not able to find any practical algorithm to do so. Nevertheless, experimentally the Fixed Mask Sampler has strong performance and beats previously proposed baselines for different classes of generative models.

## 4 Theory

As elaborated in Section 3, one of our key goals is to learn the best restoration model for the measurements at all noise levels, i.e., the function $\boldsymbol{h}(A, \boldsymbol{y}_t, t) = \mathbb{E}[\boldsymbol{x}_0|\boldsymbol{y}_t, A]$. We now show that under a certain assumption on the distribution of $A$ and $\tilde{A}$, the true population minimizer of Eq. 3.2 is indeed essentially of the form above. This assumption formalizes the notion that even conditional on $\tilde{A}$, $A$ has considerable variability, and the latter ensures that the best way to predict $A\boldsymbol{x}_0$ as a function of $\tilde{A}\boldsymbol{x}_t$ and $\tilde{A}$ is to optimally predict $\boldsymbol{x}_0$ itself. All proofs are deferred to the Appendix.

**Theorem 4.1.** *Assume a joint distribution of corruption matrices $A$ and further corruption $\tilde{A}$. If for all $\tilde{A}$ in the support it holds that $\mathbb{E}_{A|\tilde{A}}[A^T A]$ is full-rank, then the unique minimizer of the objective in equation 3.2 is given by*

$$\boldsymbol{h}_{\theta^*}(\tilde{A}, \boldsymbol{y}_t, t) = \mathbb{E}[\boldsymbol{x}_0 \mid \tilde{A}\boldsymbol{x}_t, \tilde{A}] \tag{4.1}$$

Two simple examples that fit into this framework (see Corollaries A.1 and A.2 in the Appendix) are:

- Inpainting: $A \in \mathbb{R}^{n \times n}$ is a diagonal matrix where each entry $A_{ii} \sim \text{Ber}(1-p)$ for some $p > 0$ (independently for each $i$), and the additional noise is generated by drawing $\tilde{A}|A$ such that $\tilde{A}_{ii} = A_{ii} \cdot \text{Ber}(1-\delta)$ for some small $\delta > 0$ (again independently for each $i$).[1]
- Gaussian measurements: $A \in \mathbb{R}^{m \times n}$ consists of $m$ rows drawn independently from $\mathcal{N}(0, I_n)$, and $\tilde{A} \in \mathbb{R}^{m \times n}$ is constructed conditional on $A$ by zeroing out its last row.

Notice that the minimizer in Eq 4.1 is not entirely of the form we originally desired, which was $\boldsymbol{h}(A, \boldsymbol{y}_t, t) = \mathbb{E}[\boldsymbol{x}_0 \mid A\boldsymbol{x}_t, A]$. In place of $A$, we now have $\tilde{A}$, which is a further degraded matrix.

---

[1] $\text{Ber}(q)$ indicates a Bernoulli random variable with a probability of $q$ to equal 1 and $1 - q$ for 0.

Indeed, one trivial way to satisfy the condition in Theorem 4.1 is by forming $\tilde{A}$ completely independently of $A$, e.g. by always setting $\tilde{A} = 0$. However, in this case, the function we learn is not very useful. For this reason, we would like to add as little further noise as possible and ensure that $\tilde{A}$ is close to $A$. In natural noise models such as the inpainting noise model, by letting the additional corruption probability $\delta$ approach 0, we can indeed ensure that $\tilde{A}$ follows a distribution very close to that of $A$.

## 5 Experimental Evaluation

### 5.1 Training from scratch on corrupted data

Our first experiment is to train diffusion models from scratch using corrupted training data at different levels of corruption. The corruption model we use for these experiments is random inpainting: we form our dataset by deleting each pixel with probability $p$. To create the matrix $\tilde{A}$, we further delete each row of $A$ with probability $\delta$ – this removes an additional $\delta$-fraction of the surviving pixels. Unless mentioned otherwise, we use $\delta = 0.1$. We train models on CIFAR-10, AFHQ, and CelebA-HQ. All our models are trained with corruption level $p \in \{0.0, 0.2, 0.4, 0.6, 0.8, 0.9\}$. We use the EDM [30] codebase to train our models. We replace convolutions with Gated Convolutions [64] which are known to perform better for inpainting-type problems. To use the mask $\tilde{A}$ as an additional input to the model, we simply concatenate it with the image $x$. The full training details can be found in the Appendix, Section C.

We first evaluate the restoration performance of our model for the task it was trained on (random inpainting and noise). We compare with state-of-the-art diffusion models that were trained on clean data. Specifically, for AFHQ we compare with the state-of-the-art EDM model [30] and for CelebA we compare with DDIM [56]. These models were not trained to denoise, but we can use the prior learned in the denoiser as in [60, 29] to solve any inverse problem. We experiment with the state-of-the-art reconstruction algorithms: DDRM [34] and DPS [11].

We summarize the results in Table 1. Our model performs similarly to other diffusion models, even though it has never been trained on clean data. Further, it does so by requiring only one step, while all the baseline diffusion models require hundreds of steps to solve the same task with inferior or comparable performance. The performance of DDRM improves with more function evaluations at the cost of more computation. For DPS, we did not observe significant improvement by increasing the number of steps to more than 100. We include results with noisy inpainted measurements and comparisons with a supervised method in the Appendix, Section E, Tables 3, 5. We want to emphasize that all the baselines we compare against have an advantage: they are trained on *uncorrupted* data. Instead, our models were only trained on corrupted data. This experiment indicates that: i) our training algorithm for learning the conditional expectation worked and ii) that the choice of corruption that diffusion models are trained to reverse matters for solving inverse problems.

Next, we evaluate the performance of our diffusion models as generative models. To the best of our knowledge, the only generative baseline with quantitative results for training on corrupted data is AmbientGAN [7] which is trained on CIFAR-10. We further compare with a diffusion model trained without our further corruption algorithm. We plot the results in Figure 3. The diffusion model trained without our further corruption algorithm performs well for low corruption levels but collapses entirely for high corruption. Instead, our model trained with further corruption maintains reasonable corruption scores even for high corruption levels, outperforming the previous state-of-the-art AmbientGAN for all ranges of corruption levels.

For CelebA-HQ and AFHQ we could not find any generative baselines trained on corrupted data to compare against. Nevertheless, we report FID and Inception Scores and summarize our results in Table 4 to encourage further research in this area. As shown in the Table, for CelebA-HQ and AFHQ, we manage to maintain a decent FID score even with $90\%$ of the pixels deleted. For CIFAR-10, the performance degrades faster, potentially because of the lower resolution of the training images.

### 5.2 Finetuning foundation models on corrupted data

We can apply our technique to finetune a foundational diffusion model. For all our experiments, we use Deepfloyd's IF model [2], which is one of the most powerful open-source diffusion generative

| Dataset | Corruption Probability | Method | LPIPS | PSNR | NFE |
|---------|----------------------|--------|-------|------|-----|
| CelebA-HQ | | Ours | **0.037** | **31.51** | 1 |
| | | DPS | 0.053 | 28.21 | 100 |
| | 0.6 | | 0.139 | 25.76 | 35 |
| | | DDRM | 0.088 | 27.38 | 99 |
| | | | 0.069 | 28.16 | 199 |
| | | Ours | **0.084** | **26.80** | 1 |
| | | DPS | 0.107 | 24.16 | 100 |
| | 0.8 | | 0.316 | 20.37 | 35 |
| | | DDRM | 0.188 | 22.96 | 99 |
| | | | 0.153 | 23.82 | 199 |
| | | Ours | **0.152** | **23.34** | 1 |
| | | DPS | 0.168 | 20.89 | 100 |
| | 0.9 | | 0.461 | 15.87 | 35 |
| | | DDRM | 0.332 | 18.74 | 99 |
| | | | 0.242 | 20.14 | 199 |
| AFHQ | | Ours | 0.030 | 33.27 | 1 |
| | | DPS | **0.020** | **34.06** | 100 |
| | 0.4 | | 0.122 | 25.18 | 35 |
| | | DDRM | 0.091 | 26.42 | 99 |
| | | | 0.088 | 26.52 | 199 |
| | | Ours | 0.062 | 29.46 | 1 |
| | | DPS | **0.051** | **30.03** | 100 |
| | 0.6 | | 0.246 | 20.76 | 35 |
| | | DDRM | 0.166 | 22.79 | 99 |
| | | | 0.160 | 22.93 | 199 |
| | | Ours | 0.124 | **25.37** | 1 |
| | | DPS | **0.107** | 25.30 | 100 |
| | 0.8 | | 0.525 | 14.56 | 35 |
| | | DDRM | 0.295 | 18.08 | 99 |
| | | | 0.258 | 18.86 | 199 |

Table 1: Comparison of our model (trained on corrupted data) with state-of-the-art diffusion models on CelebA (DDIM [56] model) and AFHQ (EDM [30] model) for solving the random inpainting inverse problem. Our model performs on par with state-of-the-art diffusion inverse problem solvers, even though it has never seen uncorrupted training data. Further, this is achieved with a single score function evaluation. To solve this problem with a standard pre-trained diffusion model we need to use a reconstruction algorithm (such as DPS [11] or DDRM [34]) that typically requires hundreds of steps.

models available. We choose this model over Stable Diffusion [51] because it works in the pixel space (and hence our algorithm directly applies).

**Memorization.** We show that we can finetune a foundational model on a limited dataset without memorizing the training examples. This experiment is motivated by the recent works of Carlini et al. [8], Somepalli et al. [55], and Jagielski et al. [27] that show that diffusion generative models memorize training samples and they do it significantly more than previous generative models, such as GANs, especially when the training dataset is small. Specifically, Somepalli et al. [55] train diffusion models on subsets of size $\{300, 3000, 30000\}$ of CelebA and they show that models trained on 300 or 3000 memorize and blatantly copy images from their training set.

We replicate this training experiment by finetuning the IF model on a subset of CelebA with 3000 training examples. Results are shown in Figure 1. Standard finetuning of Deepfloyd's IF on 3000 images memorizes samples and produces almost exact copies of the training set. Instead, if we corrupt the images by deleting $80\%$ of the pixels prior to training and finetune, the memorization decreases sharply and there are distinct differences between the generated images and their nearest neighbors from the dataset. This is in spite of finetuning until convergence.

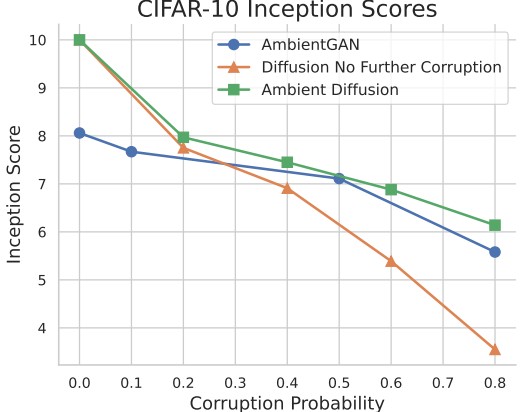

**Figure 3:** Performance on CIFAR-10 as a function of the corruption level. We compare our method with a diffusion model trained without our further corruption trick and AmbientGAN [7]. Ambient Diffusion outperforms both baselines for all ranges of corruption levels.

| Dataset | Corruption Probability | FID | Inception Score |
|---------|------------------------|-----|-----------------|
| CelebA-HQ | 0.0 | 3.26 | |
| | 0.2 | 4.18 | |
| | 0.6 | 6.08 | N/A |
| | 0.8 | 11.19 | |
| | 0.9 | 25.53 | |
| AFHQ | 0.0 | 2.41 | |
| | 0.2 | 4.47 | |
| | 0.4 | 6.96 | |
| | 0.6 | 10.11 | N/A |
| | 0.8 | 16.78 | |
| | 0.9 | 41.00 | |
| CIFAR-10 | 0.0 | 1.85 | 9.94 |
| | 0.2 | 11.70 | 7.97 |
| | 0.4 | 18.85 | 7.45 |
| | 0.6 | 28.88 | 6.88 |
| | 0.8 | 46.27 | 6.14 |

**Figure 4:** Inception/FID results on random inpainting for models trained with our algorithm on CelebA-HQ, AFHQ and CIFAR-10.

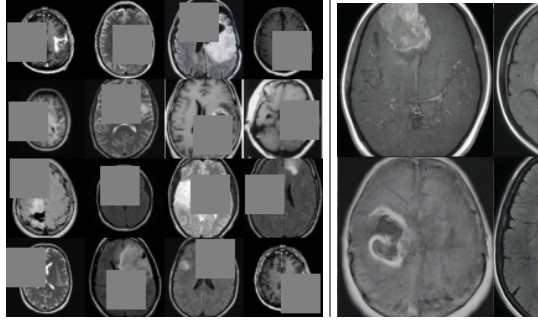

**Figure 5: Left panel:** We finetune Deepfloyd's IF diffusion model to make it a generative model for MRI images of brains with tumors. We use a small dataset [25] of only 155 images that was corrupted by removing large blocks as shown. **Right panel:** Generated samples from our finetuned model. As shown, the model learns the statistics of full brain tumor MRI images. The training set was resized to $64 \times 64$ but the generated images are at $256 \times 256$. The higher resolution is obtained by simply leveraging the power of the cascaded IF model.

To quantify the memorization, we follow the methodology of Somepalli et al. [55]. Specifically, we generate 10000 images from each model and we use DINO [9]-v2 [46] to compute top-1 similarity to the training images. Results are shown in Figure 6. Similarity values above 0.95 roughly correspond to the same person while similarities below 0.75 typically correspond to random faces. The standard finetuning (Red) often generates images that are near-identical with the training set. Instead, finetuning with corrupted samples (blue) shows a clear shift to the left. Visually we never observed a near-copy generated from our process – see also Figure 1.

We repeat this experiment for models trained on the full CelebA dataset and at different levels of corruption. We include the results in Figure 8 of the Appendix. As shown, the more we increase the corruption level the more the distribution of similarities shifts to the left, indicating less memorization. However, this comes at the cost of decreased performance, as reported in Table 4.

**New domains and different corruption.** We show that we can also finetune a pre-trained foundation model on a *new domain* given a limited-sized dataset in a few hours in a single GPU. Figure 5 shows generated samples from a finetuned model on a dataset containing 155 examples of brain tumor MRI images [25]. As shown, the model learns the statistics of full brain tumor MRI images while only trained on brain-tumor images that have a random box obfuscating $25\%$ of the image. The training set was resized to $64 \times 64$ but the generated images are at $256 \times 256$ by simply leveraging the power of the cascaded Deepfloyd IF.

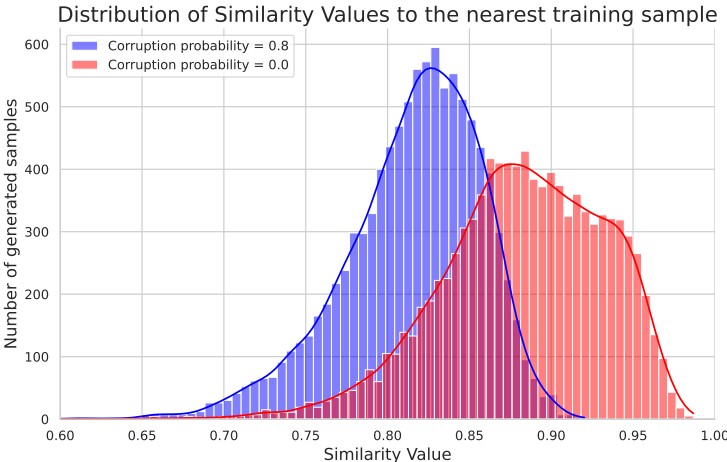

Figure 6: Distribution of similarity values to the nearest neighbor in the dataset for a finetuned IF model on a 3000 samples CelebA subset. Please note that similarity values above $0.95$ roughly correspond to the same person and similarities below $0.75$ typically correspond to random faces. Therefore the baseline finetuning process (red) often generates images that are near copies of the training set. On the contrary, our fine-tuning with corrupted samples (blue) shows a clear shift to the left. Visually we never observed a near-identical image generated from our process, see also Figure 1 for qualitative results.

**Limitations.** Our work has several limitations. First, there is a tradeoff between generator quality and corruption levels. For higher corruption, it is less likely that our generator memorizes parts of training examples, but at a cost of degrading quality. Precisely characterizing this trade-off is an open research problem. Further, in this work, we only experimented with very simple approximation algorithms to estimate $\mathbb{E}[\boldsymbol{x}_0|\boldsymbol{x}_t]$ using our trained models. Additionally, we cannot make any strict privacy claim about the protection of any training sample without making assumptions about the data distribution. We show in the Appendix that it is possible to recover $\mathbb{E}[\boldsymbol{x}_0|\boldsymbol{x}_t]$ exactly using our restoration oracle, but we do not have an algorithm to do so. Finally, our method cannot handle measurements that also have noise. Future work could potentially address this limitation by exploiting SURE regularization as in [1].

## Acknowledgments and Disclosure of Funding

The authors would like to thank Tom Goldstein for insightful discussions that benefited this work. This research has been supported by NSF Grants CCF 1763702, AF 1901292, CNS 2148141, Tripods CCF 1934932, NSF AI Institute for Foundations of Machine Learning (IFML) 2019844, the Texas Advanced Computing Center (TACC) and research gifts by Western Digital, WNCG IAP, UT Austin Machine Learning Lab (MLL), Cisco and the Archie Straiton Endowed Faculty Fellowship. Giannis Daras has been supported by the Onassis Fellowship (Scholarship ID: F ZS 012-1/2022-2023), the Bodossaki Fellowship and the Leventis Fellowship. Aravind Gollakota was at UT Austin while this work was done.

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

# A Proofs

*Proof of Theorem 4.1.* Let $\boldsymbol{h}_{\theta^*}$ be a minimizer of equation 3.2, and for brevity let

$$\boldsymbol{f}(\tilde{A}\boldsymbol{x}_t, \tilde{A}) = \boldsymbol{h}_{\theta^*}(\tilde{A}\boldsymbol{x}_t, \tilde{A}) - \mathbb{E}[\boldsymbol{x}_0 \mid \tilde{A}\boldsymbol{x}_t, \tilde{A}]$$

be the difference between $\boldsymbol{h}_{\theta^*}$ and the claimed optimal solution. We will now argue that $\boldsymbol{f}$ must be identically zero.

First, by adding and subtracting $A\mathbb{E}[\boldsymbol{x}_0 \mid \tilde{A}\boldsymbol{x}_t, \tilde{A}]$, we can expand the objective value achieved at $\theta = \theta^*$ in equation 3.2 as follows:

$$J^{\mathrm{corr}}(\theta^*) = \mathbb{E}_{\boldsymbol{x}_0, \boldsymbol{x}_t, A, \tilde{A}} \left[ \left\| (A\boldsymbol{x}_0 - A\mathbb{E}[\boldsymbol{x}_0 \mid \tilde{A}\boldsymbol{x}_t, \tilde{A}]) - A\boldsymbol{f}(\tilde{A}\boldsymbol{x}_t, \tilde{A}) \right\|^2 \right]$$

$$= \mathbb{E}_{\boldsymbol{x}_0, \boldsymbol{x}_t, A, \tilde{A}} \left[ \left\| A\boldsymbol{x}_0 - A\mathbb{E}[\boldsymbol{x}_0 \mid \tilde{A}\boldsymbol{x}_t, \tilde{A}] \right\|^2 \right] + \mathbb{E}_{\boldsymbol{x}_0, \boldsymbol{x}_t, A, \tilde{A}} \left[ \left\| A\boldsymbol{f}(\tilde{A}\boldsymbol{x}_t, \tilde{A}) \right\|^2 \right]$$

$$- 2\mathbb{E}_{\boldsymbol{x}_0, \boldsymbol{x}_t, A, \tilde{A}} \left[ (A\boldsymbol{x}_0 - A\mathbb{E}[\boldsymbol{x}_0 \mid \tilde{A}\boldsymbol{x}_t, \tilde{A}])^T A\boldsymbol{f}(\tilde{A}\boldsymbol{x}_t, \tilde{A}) \right].$$

Here the first term is the irreducible error, while the third term vanishes by the tower law of expectations:

$$\mathbb{E}_{\boldsymbol{x}_0, \boldsymbol{x}_t, A, \tilde{A}} \left[ (A\boldsymbol{x}_0 - A\mathbb{E}[\boldsymbol{x}_0 \mid \tilde{A}\boldsymbol{x}_t, \tilde{A}])^T A\boldsymbol{f}(\tilde{A}\boldsymbol{x}_t, \tilde{A}) \right]$$

$$= \mathbb{E}_{\boldsymbol{x}_0, A, \tilde{A}, \tilde{A}\boldsymbol{x}_t} \left[ (\boldsymbol{x}_0 - \mathbb{E}[\boldsymbol{x}_0 \mid \tilde{A}\boldsymbol{x}_t, \tilde{A}])^T A^T A\boldsymbol{f}(\tilde{A}\boldsymbol{x}_t, \tilde{A}) \right]$$

$$= \mathbb{E}_{A, \tilde{A}, \tilde{A}\boldsymbol{x}_t} \left[ \mathbb{E}_{\boldsymbol{x}_0 \mid A, \tilde{A}, \tilde{A}\boldsymbol{x}_t} \left[ \boldsymbol{x}_0 - \mathbb{E}[\boldsymbol{x}_0 \mid \tilde{A}\boldsymbol{x}_t, \tilde{A}] \right]^T A^T A\boldsymbol{f}(\tilde{A}\boldsymbol{x}_t, \tilde{A}) \right]$$

$$= \mathbb{E}_{A, \tilde{A}, \tilde{A}\boldsymbol{x}_t} \left[ (\mathbb{E}[\boldsymbol{x}_0 \mid \tilde{A}\boldsymbol{x}_t, \tilde{A}] - \mathbb{E}[\boldsymbol{x}_0 \mid \tilde{A}\boldsymbol{x}_t, \tilde{A}])^T A^T A\boldsymbol{f}(\tilde{A}\boldsymbol{x}_t, \tilde{A}) \right]$$

$$= 0.$$

Thus the only part of $J^{\mathrm{corr}}(\theta^*)$ that actually depends on the parameter value $\theta^*$ is the second term. We now show that the second term can be made to vanish, and that this occurs precisely when $\boldsymbol{f}$ is identically 0:

$$\mathbb{E}_{\boldsymbol{x}_0, \boldsymbol{x}_t, A, \tilde{A}} \left[ \left\| A\boldsymbol{f}(\tilde{A}\boldsymbol{x}_t, \tilde{A}) \right\|^2 \right]$$

$$= \mathbb{E}_{\boldsymbol{x}_0, \boldsymbol{x}_t, A, \tilde{A}} \left[ \boldsymbol{f}(\tilde{A}\boldsymbol{x}_t, \tilde{A})^T A^T A\boldsymbol{f}(\tilde{A}\boldsymbol{x}_t, \tilde{A}) \right]$$

$$= \mathbb{E}_{\boldsymbol{x}_0, \boldsymbol{x}_t, \tilde{A}} \left[ \boldsymbol{f}(\tilde{A}\boldsymbol{x}_t, \tilde{A})^T \mathbb{E}_{A \mid \boldsymbol{x}_0, \boldsymbol{x}_t, \tilde{A}} \left[ A^T A \right] f(\tilde{A}\boldsymbol{x}_t, \tilde{A}) \right]$$

$$= \mathbb{E}_{\boldsymbol{x}_0, \boldsymbol{x}_t, \tilde{A}} \left[ f(\tilde{A}\boldsymbol{x}_t, \tilde{A})^T \mathbb{E}_{A \mid \tilde{A}} \left[ A^T A \right] \boldsymbol{f}(\tilde{A}\boldsymbol{x}_t, \tilde{A}) \right].$$

For every $\tilde{A}$ and $\boldsymbol{x}_0, \boldsymbol{x}_t$, by assumption we have that $\mathbb{E}_{A \mid \tilde{A}} \left[ A^T A \right]$ is full-rank, and so the inner quadratic form is minimized when $\boldsymbol{f}(\tilde{A}\boldsymbol{x}_t, \tilde{A}) = 0$. Further, the term as a whole vanishes exactly when this holds for every $\tilde{A}$ and $\boldsymbol{x}_0, \boldsymbol{x}_t$ in the support, which means $\boldsymbol{f}$ must be identically zero. $\qquad\square$

**Corollary A.1** (Inpainting noise model). *Consider the following inpainting noise model: $A \in \mathbb{R}^{n \times n}$ is a diagonal matrix where each entry $A_{ii} \sim \mathrm{Ber}(1 - p)$ for some $p > 0$ (independently for each $i$), and the additional noise is generated by drawing $\tilde{A} \mid A$ such that $\tilde{A}_{ii} = A_{ii} \mathrm{Ber}(1 - \delta)$ for some small $\delta > 0$ (again independently for each $i$). Then the unique minimizer of the objective in equation 3.2 is*

$$\boldsymbol{h}_{\theta^*}(\tilde{A}\boldsymbol{x}_t, \tilde{A}) = \mathbb{E}[\boldsymbol{x}_0 \mid \tilde{A}\boldsymbol{x}_t, \tilde{A}].$$

*Proof.* By Theorem 4.1, what we must show is that for any $\tilde{A}$ in the support, $E_{A \mid \tilde{A}}[A^T A]$ is full-rank. Fix any particular realization of $\tilde{A}$, which will be a diagonal matrix with only 0s and 1s. For indices $i$ where $\tilde{A}_{ii} = 1$, we know that for any $A$ drawn conditional on $\tilde{A}$, $A_{ii} = 1$ as well, i.e.

$$\Pr(A_{ii} = 1 \mid \tilde{A}_{ii} = 1) = 1.$$

And for indices $i$ where $\tilde{A}_{ii} = 0$, by Bayes' rule we have

$$\Pr(A_{ii} = 1 \mid \tilde{A}_{ii} = 0) = \frac{\Pr(A_{ii} = 1, \tilde{A}_{ii} = 0)}{\Pr(A_{ii} = 0, \tilde{A}_{ii} = 0) + \Pr(A_{ii} = 1, \tilde{A}_{ii} = 0)} = \frac{(1-p)\delta}{(1-p)\delta + p} =: q.$$

Thus we see that $E_{A|\tilde{A}}[A^T A] = E_{A|\tilde{A}}[A]$ is a diagonal matrix whose entries are 1 wherever $\tilde{A}_{ii} = 1$ and $q$ wherever $\tilde{A}_{ii} = 0$. This is clearly of full rank since $q > 0$. $\qquad\square$

**Corollary A.2** (Gaussian measurements noise model). *Consider the following noise model where we only observe $m$ independent Gaussian measurements of the ground truth, and then one of the measurements is further omitted: $A \in \mathbb{R}^{m \times n}$ consists of $m$ rows drawn independently from $\mathcal{N}(0, I_n)$, and $\tilde{A} \in \mathbb{R}^{m \times n}$ is constructed condtional on $A$ by zeroing out its last row. Then the unique minimizer of the objective in equation 3.2 is*

$$\boldsymbol{h}_{\theta^*}(\tilde{A}\boldsymbol{x}_t, \tilde{A}) = \mathbb{E}[\boldsymbol{x}_0 \mid \tilde{A}\boldsymbol{x}_t, \tilde{A}].$$

*Proof.* Again by Theorem 4.1, we must show that for any $\tilde{A}$ in the support, $E_{A|\tilde{A}}[A^T A]$ is full-rank. Fix any realization of $\tilde{A}$, which will have the following form:

$$\tilde{A} = \begin{bmatrix} \rule{0.6cm}{0.4pt} & \boldsymbol{a}_1^T & \rule{0.6cm}{0.4pt} \\ & \vdots & \\ \rule{0.6cm}{0.4pt} & \boldsymbol{a}_{m-1}^T & \rule{0.6cm}{0.4pt} \\ \rule{0.6cm}{0.4pt} & 0^T & \rule{0.6cm}{0.4pt} \end{bmatrix}.$$

Then it is clear that conditional on $\tilde{A}$, $A$ has the following distribution:

$$A \mid \tilde{A} = \begin{bmatrix} \rule{0.6cm}{0.4pt} & \boldsymbol{a}_1^T & \rule{0.6cm}{0.4pt} \\ & \vdots & \\ \rule{0.6cm}{0.4pt} & \boldsymbol{a}_{m-1}^T & \rule{0.6cm}{0.4pt} \\ \rule{0.6cm}{0.4pt} & \boldsymbol{b}_m^T & \rule{0.6cm}{0.4pt} \end{bmatrix} \quad \text{where} \quad \boldsymbol{b}_m \sim \mathcal{N}(0, I_n).$$

Here $\boldsymbol{b}_m$ is drawn entirely independently from $\mathcal{N}(0, I_n)$. Elementary manipulations now reveal that $\mathbb{E}_{A|\tilde{A}}[A^T A] = \mathbb{E}_{\boldsymbol{b}_m \sim \mathcal{N}(0,I_n)}[\tilde{A}^T\tilde{A} + \boldsymbol{b}_m\boldsymbol{b}_m^T] = \tilde{A}^T\tilde{A} + I_n$, which is clearly full rank (indeed, it is PSD with strictly positive eigenvalues). $\qquad\square$

## A.1 Reduction

In this section we argue that if there is an algorithm that recovers the target distribution $p_0(\boldsymbol{x}_0)$ from i.i.d. samples $(A\boldsymbol{x}_0, A)$ where $A \sim p(A)$ and $\boldsymbol{x}_0 \sim p_0(\boldsymbol{x}_0)$, then, there is an algorithm that recovers $p_0(\boldsymbol{x}_0)$ without sample access, but instead, using access to an oracle that given $t, \boldsymbol{x}$ and $A$ in the support of $p(A)$, returns $\mathbb{E}[\boldsymbol{x}_0 \mid A\boldsymbol{x}_t]$.

Indeed, for any $A$, Chen et al. [10] show that it is possible to recover the distribution of $A\boldsymbol{x}_0$ given access to $\mathbb{E}[A\boldsymbol{x}_0 \mid A\boldsymbol{x}_t]$ for any $t$ and $\boldsymbol{x}$ under some minimal assumptions on the data distribution $p_0(\boldsymbol{x}_0)$, see [10, Assumptions 1-3]. Using our oracle and using this theorem, we can recover the distribution of $A\boldsymbol{x}_0$ for all $A$ in the support. By sampling from these distributions, one can as well obtain samples of $A\boldsymbol{x}_0$ for $A \sim p(A)$ and $\boldsymbol{x}_0 \sim p_0(\boldsymbol{x}_0)$. Hence, if these samples are sufficient to recover $p_0(\boldsymbol{x}_0)$, then, having an oracle to these conditional expectations is sufficient as well.

This intuition can be formalized as follows. Fix a distribution $p_A(A)$ over corruption matrices. For a distribution $p_0(\boldsymbol{x}_0)$, denote by $\text{corrupt}(p_A, p_0)$ the distribution over pairs $(A, A\boldsymbol{x}_0)$ where $A \sim p_A(A)$ and $\boldsymbol{x}_0 \sim p_0(\boldsymbol{x}_0)$. We say that *it is possible to reconstruct $p_0(\boldsymbol{x}_0)$ from random corruptions $A \sim p_A(A)$* if the following holds: for any two distributions, $p_0(\boldsymbol{x}_0)$ and $p_0'(\boldsymbol{x}_0')$ that satisfy Assumptions 1-3 of Chen et al. [10], if $\text{corrupt}(p_A, p_0) = \text{corrupt}(p_A, p_0')$, then $p_0 = p_0'$. Similarly, we say that *it is possible to reconstruct $p_0(\boldsymbol{x}_0)$ from conditional expectations given $A \sim p(A)$* if the following holds: for any distribution $p_0(\boldsymbol{x}_0)$ and $p_0'(\boldsymbol{x}_0')$ that satisfy Assumptions 1-3 of Chen et al. [10], if for all $x$, $t$ and $A$ in the support of $p_A$,

$$\mathbb{E}_{(\boldsymbol{x}_0,\boldsymbol{x}_t)\sim p_{0,t}(\boldsymbol{x}_0,\boldsymbol{x}_t)}[\boldsymbol{x}_0 \mid A\boldsymbol{x}_t = \boldsymbol{x}] = \mathbb{E}_{(\boldsymbol{x}_0',\boldsymbol{x}_t')\sim p_{0,t}'(\boldsymbol{x}_0',\boldsymbol{x}_t')}[\boldsymbol{x}_0' \mid A\boldsymbol{x}_t' = \boldsymbol{x}] \tag{A.1}$$

then $p_0 = p_0'$. Here, $p_{0,t}(\boldsymbol{x}_0, \boldsymbol{x}_t)$ is obtained by sampling $\boldsymbol{x}_0 \sim p_0$ and $\boldsymbol{x}_t = \boldsymbol{x}_0 + \sigma_t \boldsymbol{\eta}$ where $\boldsymbol{\eta} \sim \mathcal{N}(0, I)$. Similarly, $p_{0,t}'(\boldsymbol{x}_0', \boldsymbol{x}_t')$ is obtained by the same process where $\boldsymbol{x}_0'$ is instead sampled from $p_0'$. We state the following lemma:

**Lemma A.3.** *Fix a distribution $p_A(A)$. If it is possible to reconstruct $p_0(\boldsymbol{x}_0)$ from random corruptions $\boldsymbol{y}_0 = A\boldsymbol{x}_0$, $A \sim p_A(A)$, then it is possible to reconstruct $p_0(\boldsymbol{x}_0)$ given access to an oracle that computes the conditional expectations $\mathbb{E}[\boldsymbol{x}_0 | A\boldsymbol{x}_t, A]$, for $A \sim p_A(A)$ and $\boldsymbol{x}_t = \boldsymbol{x}_0 + \sigma_t \boldsymbol{\eta}$.*

*Proof.* Assume that it is possible to reconstruct $p_0(\boldsymbol{x}_0)$ from random corruptions $A \sim p_A(A)$ and we will prove that it is possible to reconstruct $p_0(\boldsymbol{x}_0)$ from conditional expectations given $A \sim p_A(A)$. To do so, let $p_0(\boldsymbol{x}_0)$ and $p_0'(\boldsymbol{x}_0')$ be two distributions that satisfy Assumptions 1-3 of Chen et al. [10]. Assume that equation A.1 holds. We will prove that $p_0 = p_0'$. Fix some $A$ in the support of $p_A$. By Chen et al. [10], there is an algorithm that samples from the distribution of $A\boldsymbol{x}_0$, $\boldsymbol{x}_0 \sim p_0(\boldsymbol{x}_0)$, that only has access to $\mathbb{E}[A\boldsymbol{x}_0 \mid A\boldsymbol{x}_t, A]$ and similarly, there is an algorithm that samples from the distribution of $A\boldsymbol{x}_0'$ that only has access to $\mathbb{E}[A\boldsymbol{x}_0' \mid A\boldsymbol{x}_t', A]$. Since these two conditional expectations are assumed to be the same, then the distribution of $A\boldsymbol{x}_0$ equals the distribution of $A\boldsymbol{x}_0'$. Consequently, $\text{corrupt}(p_A, p_0) = \text{corrupt}(p_A, p_0')$. By the assumption that it is possible to reconstruct $p_0(\boldsymbol{x}_0)$ from random corruptions $A \sim p_A(A)$, this implies that $p_0 = p_0'$. This completes the proof. $\qquad\square$

## B  Broader Impact and Risks

Generative models in general hold the potential to have far-reaching impacts on society in a variety of forms, coupled with several associated risks [43, 32, 33, 31]. Among other potential applications, they can be utilized to create deceptive images and perpetuate societal biases. To the best of our knowledge, our paper does not amplify any of these existing risks. Regarding the included MRI results, we want to clarify that we make no claim that such results are diagnostically useful. This experiment serves only as a toy demonstration that it can be potentially feasible to learn the distribution of MRI scans with corrupted samples. Significant further research must be done in collaboration with radiologists before our algorithm gets tested in clinical trials. Finally, we want to underline again that even though our approach seems to mitigate the memorization issue in generative models, we cannot guarantee the privacy of any training sample unless we make assumptions about the data distribution. Hence, we strongly discourage using this algorithm in applications where privacy is important before this research topic is investigated further.

## C  Training Details

We open-source our code and models to facilitate further research in this area: https://github.com/giannisdaras/ambient-diffusion.

For all the experiments in this paper, we concatenated the image with the mask along the channel dimension to condition the network on the measurement operator, i.e. the inpainting mask is an additional input channel.

**Models trained from scratch.**  We trained models from scratch at different corruption levels on CelebA-HQ, AFHQ and CIFAR-10. The resolution of the first two datasets was set to $64 \times 64$ and for CIFAR-10 we trained on $32 \times 32$.

We started from EDM's [30] official implementation and made some necessary changes. Architecturally, the only change we made was to replace the convolutional layers with Gated Convolutions [64] that are known to perform well for inpainting problems. We observed that this change stabilized the training significantly, especially in the high-corruptions regime. As in EDM, we use the architecture from the DDPM++ [37] paper.

To avoid additional design complexity, we tried to keep our hyperparameters as close as possible to the EDM paper. We observed that for high corruption levels, it was useful to add gradient clipping, otherwise, the training would often diverge. For all our experiments, we use gradient clipping with max-norm set to 1.0. We underline that unfortunately, even with gradient clipping, the training at high corruption levels ($p \geq 0.8$), still diverges sometimes. Whenever this happened, we restarted the training from an earlier checkpoint. We list the rest of the hyperparameters we used in Table 2.

Table 2: Training Hyperparameters

| Dataset | Iters | Batch | LR | SDE | $p$ | $\delta$ | Aug. Prob. | Ch. Multipliers | Dropout |
|---|---|---|---|---|---|---|---|---|---|
| CIFAR-10 | | 512 | 1e-3 | | $\{0.2, 0.4, 0.6, 0.8\}$ | | 0.12 | (1, 1, 1, 1) | 0.13 |
| AFHQ | 200000 | | | VP | $\{0.2, 0.4, 0.6, 0.8, 0.9\}$ | 0.1 | | | 0.25 |
| CelebA-HQ | | 256 | 2e-4 | | $\{0.2, 0.6, 0.8\}$ 0.9 | 0.3 | 0.15 | (1, 2, 2, 2) | 0.1 |

Training diffusion models from scratch is quite computationally intensive. We trained all our models for 200000 iterations. Our CIFAR-10 models required $\approx 2$ days of training each on 6 A100 GPUs. Our AFHQ and CelebA-HQ models required $\approx 6$ days of training each on 6 A100 GPUs. These numbers roughly match the performance reported in the EDM paper, indicating that the extra corruption we need to do on matrix $A$ does not increase training time.

Due to the increased computational complexity of training these models, we could not extensively optimize the hyperparameters, e.g. the $\delta$ probability in our extra corruption. For higher corruption, e.g. for $p = 0.9$, we noticed that we had to increase $\delta$ in order for the model to learn to perform well on the unobserved pixels. For a small-scale ablation study on CIFAR-10, see Table 7.

**Finetuning Deepfloyd's IF.** We access Deepfloyd's IF [2] model through the `diffusers` library. The model is a Cascaded Diffusion Model [24]. The first part of the pipeline is a text-conditional diffusion model that outputs images at resolution $64 \times 64$. Next in the pipeline, there are two diffusion models that are conditioned both in the input text and the low-resolution output of the previous stage. The first upscaling module increases the resolution from $64 \times 64$ to $256 \times 256$ and the final one from $256 \times 256$ to $1024 \times 1024$.

To reduce the computational requirements of the finetuning, we only finetune the first text-conditional diffusion model that works with $64 \times 64$ resolution images. Once the finetuning is completed, we use again the whole model to generate high-resolution images.

For the finetuning, we set the training batch size to 32 and the learning rate to $3e - 6$. We train for a maximum of 15000 steps and we keep the checkpoint that gives the lowest error on the pixels that we further corrupted. To further reduce the computational requirements, we use an 8-bit Adam Optimizer and we train with half-precision.

For our CelebA finetuning experiments, we set $\delta = 0.1$ and $p = 0.8$. We experiment with the full training set, a subset of size 3000 (see Figure 1) and a subset of 300. For the model trained with only 300 heavily corrupted samples, we did not observe memorization but the samples were of very low quality. Intuitively, our algorithm provides a way to control the trade-off between memorization and fidelity. Fully exploring this trade-off is a very promising direction for future work. For our MRI experiments, we use two non-overlapping blocks that each obfuscates 25% of the image and we evaluate the model in one of them.

All of our fine-tuning experiments can be completed in a few hours. Training for 15000 iterations takes $\approx 10$ hours on an A100 GPU, but we usually get the best checkpoints earlier in the training.

## D    Evaluation Details

**FID evaluation.** Our FID [22] score is computed with respect to the training set, as is standard practice, e.g. see [23, 30, 58]. For each of our models trained from scratch, we generate 50000 images using the seeds $0 - 49999$. Once we generate our images, we use the code provided in the official implementation of the EDM [30] paper for the FID computation.

## E    Additional Experiments

### E.1    Restoration performance with noisy measurements

In Table 1 of the main paper, we compare the restoration performance of our models and vanilla diffusion models (trained with uncorrupted images). We compare the restoration performance in the task of random inpainting because it is straightforward to use our models to solve this inverse problem.

It is potentially feasible to use our trained generative models to solve any (linear or non-linear) inverse problem but we leave this direction for future work.

In this section, we further compare our models (trained with randomly inpainted images) in the task of inpainting with noisy measurements. Concretely, we want to predict $x_0$, given measurements $y_t = A(x_0 + \sigma_{y_t}\eta)$, $\eta \sim \mathcal{N}(0, I)$. As in the rest of the paper, we assume that the mask matrix $A$ is known. We can solve this problem with one model prediction using our trained models since according to Eq. 4.1, we are learning: $h_\theta(y_t, A, t) = \mathbb{E}[x_0 | Ax_t = y_t, A]$.

We use the EDM [30] state-of-the-art model trained on AFHQ as our baseline (as we did in the main paper). To use this pre-trained model to solve the noisy random inpainting inverse problem, we need a reconstruction algorithm. We experiment again with DPS and DDRM which can both handle inverse problems with noise in the measurements. We present our results in Table 3. As shown, our models significantly outperform the EDM models that use the DDRM reconstruction algorithm and perform on-par with the EDM models that use the DPS reconstruction algorithm.

| Dataset | Corruption Probability | Measurement Noise ($\sigma_{y_0}$) | Method | LPIPS | PSNR | NFE |
|---|---|---|---|---|---|---|
| AFHQ | | | Ours | 0.0861 | 29.46 | 1 |
| | | | DPS | **0.0846** | **29.83** | 100 |
| | 0.4 | 0.05 | | 0.2061 | 24.47 | 35 |
| | | | DDRM | 0.1739 | 25.38 | 99 |
| | | | | 0.1677 | 25.58 | 199 |
| | | | Ours | 0.1031 | 27.40 | 1 |
| | | | DPS | **0.0949** | **27.92** | 100 |
| | 0.6 | 0.05 | | 0.4066 | 18.73 | 35 |
| | | | DDRM | 0.3626 | 19.49 | 99 |
| | | | | 0.3506 | 19.70 | 199 |
| | | | Ours | 0.1792 | **23.21** | 1 |
| | | | DPS | **0.1778** | 23.01 | 100 |
| | 0.8 | 0.05 | | 0.5879 | 13.65 | 35 |
| | | | DDRM | 0.5802 | 13.99 | 99 |
| | | | | 0.5753 | 14.09 | 199 |

Table 3: Comparison of our model (trained on corrupted data) with state-of-the-art diffusion models on CelebA (DDIM [56] model) and AFHQ (EDM [30] model) for solving the random inpainting inverse problem with **noise**.

## E.2 Additional Comparisons

We incude additional comparisons on CIFAR-10 and CelebA with MisGAN [40]. The MisGAN work generalizes AmbientGAN in the case where the measurement operator is unknown. Specifically, the authors propose an additional generator that learns to model the corruption mechanism with adversarial training. The results are shown in Table 4.

## E.3 Comparison with Supervised Methods

For completeness, we include a comparison with Masked AutoEncoders [21], a state-of-the-art supervised method for solving the random inpainting problem. The official repository of this paper does not include models trained on AFHQ. We compare with the available models that are trained on the iNaturalist dataset which is the most semantically close dataset we could find. We emphasize that this model was trained with access to uncorrupted images. Results are shown in 5. As shown, our method and DPS outperform this supervised baseline. We underline that this experiment is included for completeness and does not exclude the possibility that there are more performant supervised alternatives for random inpainting.

## E.4 Sampler Ablation

By Lemma A.3, if it is possible to learn $p_0(x_0)$ using corrupted samples then it is also possible to use our learned model to sample from $p_0(x_0)$. Even though such an algorithm exists, we do not know which one it is.

| CelebA | | |
|---|---|---|
| Corruption Probability | Method | FID |
| 0.6 | MisGAN | 37.42 |
| 0.6 | Ambient Diffusion | **6.08** |
| 0.8 | MisGAN | 100.0 |
| 0.8 | Ambient Diffusion | **11.19** |
| 0.9 | MisGAN | 141.11 |
| 0.9 | Ambient Diffusion | **25.53** |

| CIFAR-10 | | |
|---|---|---|
| Corruption Probability | Method | FID |
| 0.4 | MisGAN | 18.95 |
| 0.4 | Ambient Diffusion | **18.85** |
| 0.6 | MisGAN | 49.30 |
| 0.6 | Ambient Diffusion | **28.88** |
| 0.8 | MisGAN | 111.50 |
| 0.8 | Ambient Diffusion | **46.27** |

Table 4: Comparison with MisGAN.

| Corruption Probability | Method | LPIPS | PSNR | NFE |
|---|---|---|---|---|
| | Ours | 0.0304 | 33.27 | 1 |
| 0.4 | DPS | **0.0203** | **34.06** | 100 |
| | MAE | 0.0752 | 28.88 | 1 |
| | Ours | 0.0628 | 29.46 | 1 |
| 0.6 | DPS | **0.0518** | **30.03** | 100 |
| | MAE | 0.0995 | 25.89 | 1 |
| | Ours | 0.1245 | **25.37** | 1 |
| 0.8 | DPS | **0.1078** | 25.30 | 100 |
| | MAE | 0.1794 | 22.01 | 1 |

Table 5: Comparison with the MAE [21], a state-of-the-art supervised method for solving the restoration task of random inpainting.

In the paper, we proposed a simple idea for sampling, the Fixed Mask Sampler. The Fixed Mask Sampler fixes a mask throughout the sampling process. Hence, sampling with this algorithm is equivalent to first sampling some pixels from the marginals of $p_0(\boldsymbol{x}_0)$ and then completing the rest of the pixels with the best reconstruction (the conditional expectation) in the last step. This simple sampler performs remarkably well and we use it throughout the main paper.

**Sampling with Reconstruction Guidance.** In the Fixed Mask Sampler, at any point the prediction is a convex combination of the current value and the predicted denoised image. As $t \to 0$, $\gamma_t \to 0$. Hence, for the masked pixels, the fixed mask sampler outputs the conditional expectation of their value given the observed pixels. This leads to averaging effects as the corruption gets higher. To correct for this problem, we add one more term in the update: the Reconstruction Guidance term. The issue with the previous sampler is that the model never sees certain pixels. We would like to evaluate the model using different masks. However, the model outputs for the denoised image might be very different when evaluated with different masks. To account for this problem, we add an additional term that enforces updates that lead to consistency on the reconstructed image. The update of the sampler with Reconstruction Guidance becomes:

$$\boldsymbol{x}_{t-\Delta t} = \underbrace{\frac{\sigma_{t-\Delta t}}{\sigma_t}}_{\gamma_t} \boldsymbol{x}_t + \underbrace{\frac{\sigma_t - \sigma_{t-\Delta t}}{\sigma_t}}_{1-\gamma_t} \mathbb{E}[\boldsymbol{x}_0|\tilde{A}\boldsymbol{x}_t, \tilde{A}] - \frac{w_t}{2} \nabla_{\boldsymbol{x}_t} \mathbb{E}_{A' \sim p(\tilde{A}')} \left[ ||\mathbb{E}[\boldsymbol{x}_0|\tilde{A}\boldsymbol{x}_t, \tilde{A}] - \mathbb{E}[\boldsymbol{x}_0|\tilde{A}'\boldsymbol{x}_t, \tilde{A}']||^2 \right].$$

(E.1)

This sampler is inspired by the Reconstruction Guidance term used in Imagen [53]. In Table 6 we ablate the performance of this alternative sampler. For the reconstruction guidance sampler, we select each time four masks at random and we add an extra update term to the Fixed Mask Sampler that ensures that the predictions of the Fixed Mask Sampler are not very different compared to the predictions with the other four masks (that have different context regarding the current iterate $x_t$). We set the guidance parameter $w_t$ to the value $5e-4$. As shown, this sampler improves the performance, especially for the low corruption probabilities where the extra masks give significant information about the current state to the predictions given only one fixed mask. However, the benefits of this sampler are vanishing for higher corruption. Given that the two samples perform on par and that the Reconstruction Guidance Sampler is much more computationally intensive (we need one extra prediction per step for each extra mask), we choose to use the Fixed Mask Sampler for all the experiments in the paper.

| Sampler Type | Corruption Probability | FID | Inception Score |
|---|---|---|---|
| Fixed Mask | 0.2 | 11.70 | 7.97 |
| | 0.4 | 18.85 | 7.45 |
| | 0.6 | **28.88** | 6.88 |
| | 0.8 | **46.27** | **6.14** |
| Reconstruction Guidance | 0.2 | **11.59** | **8.01** |
| | 0.4 | **18.52** | **7.51** |
| | 0.6 | 28.90 | **6.91** |
| | 0.8 | 46.31 | 6.13 |

Table 6: Comparison between the Fixed Mask sampler and the Reconstruction Guidance Sampler.

## E.5 $\delta$ Ablation

A hyperparameter of our method is $\delta$, which controls the strength of the additional corruption. For the random inpainting case, $\delta$ needs to be positive in order to learn to predict the clean image for the missing pixels. On the other hand, as $\delta$ increases, so does the difficulty of the prediction task (which finally controls generation performance). In Table 7 we show how performance changes on CIFAR-10 for different values of $\delta$. The recommended setting is $\delta = 0.1$ but hyperparameter search for the value of $\delta$. might lead to performance improvements.

| $p$ | $\delta$ | Inception Score |
|---|---|---|
| 0.4 | 0.0 | 6.70 (from Figure 5) |
| 0.4 | 0.1 | 7.45 (from Figure 6) |
| 0.4 | 0.4 | 6.95 (experiment added for the rebuttal) |

Table 7: Ablation Study for the value of $\delta$.

## E.6 Additional Figures

Figure 7 shows reconstructions of AFHQ corrupted images with the EDM AFHQ model trained on clean data (columns 3, 4) and our model trained on corrupted data (column 5). The restoration task is random inpainting at probability $p = 0.8$. The last two rows also have measurement noise with $\sigma_{y_0} = 0.05$.

In Figure 8, we repeat the experiment of Figure 6 of the main paper but for models trained on the full CelebA dataset and at different levels of corruption. As shown, increasing the corruption level leads to a clear shift of the distribution to the left, indicating less memorization. This comes at the cost of decreased performance, as reported in Table 4.

| Input | Ground Truth | DDRM [34] | DPS [11] | Ours |
|-------|--------------|-----------|----------|------|

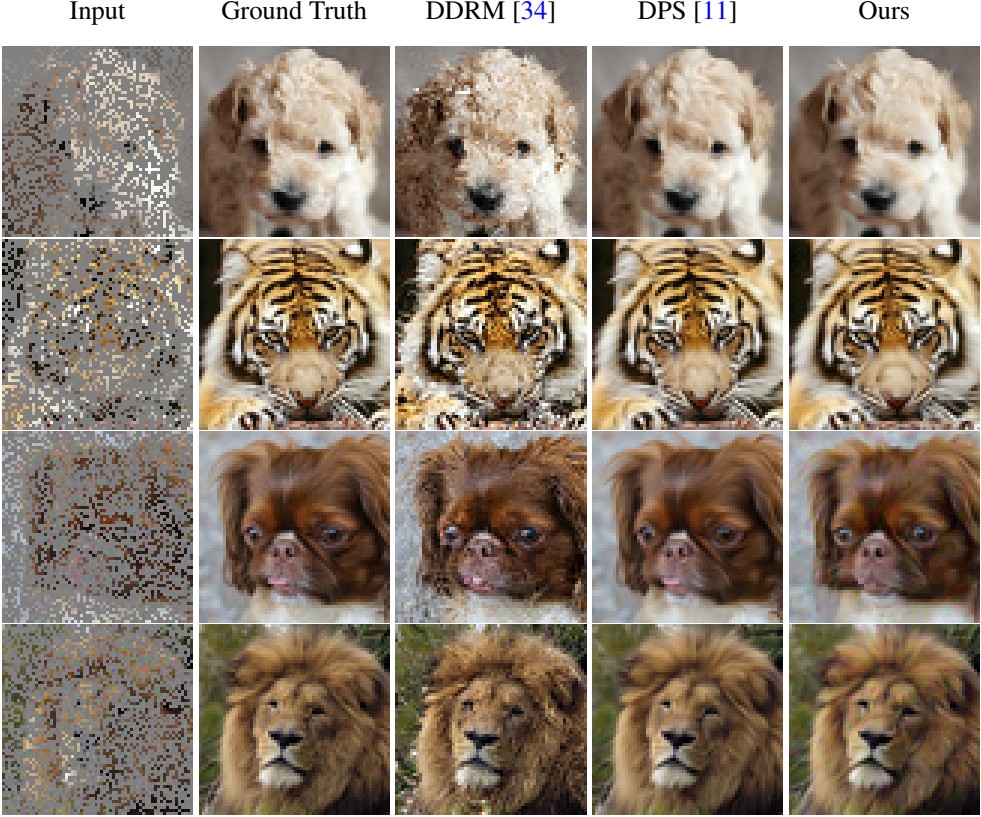

Figure 7: Reconstructions of AFHQ corrupted images with the EDM AFHQ model trained on clean data (columns 3, 4) and our model trained on corrupted data (column 5). The restoration task is random inpainting at probability $p = 0.6$. The last two rows also have measurement noise with $\sigma_{\boldsymbol{y}_0} = 0.05$.

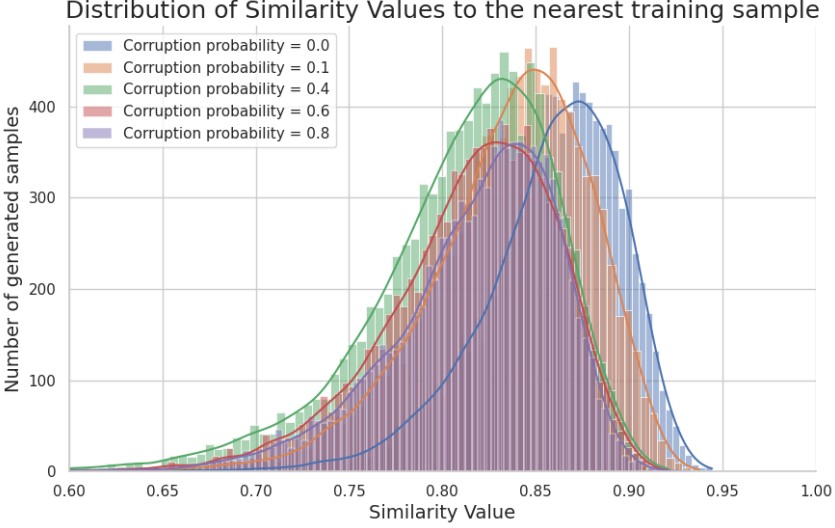

Figure 8: Distribution of similarity values to the nearest neighbor in the dataset for finetuned IF models on the full CelebA dataset at different levels of corruption. Please note that similarity values above $0.95$ roughly correspond to the same person, while similarities below $0.75$ correspond to almost random faces. As shown, increasing the corruption level leads to a clear shift of the distribution to the left, indicating less memorization.

In the remaining pages, we include uncurated unconditional generations of our models trained at different corruption levels $p$. Results are shown in Figures 9,10,11.

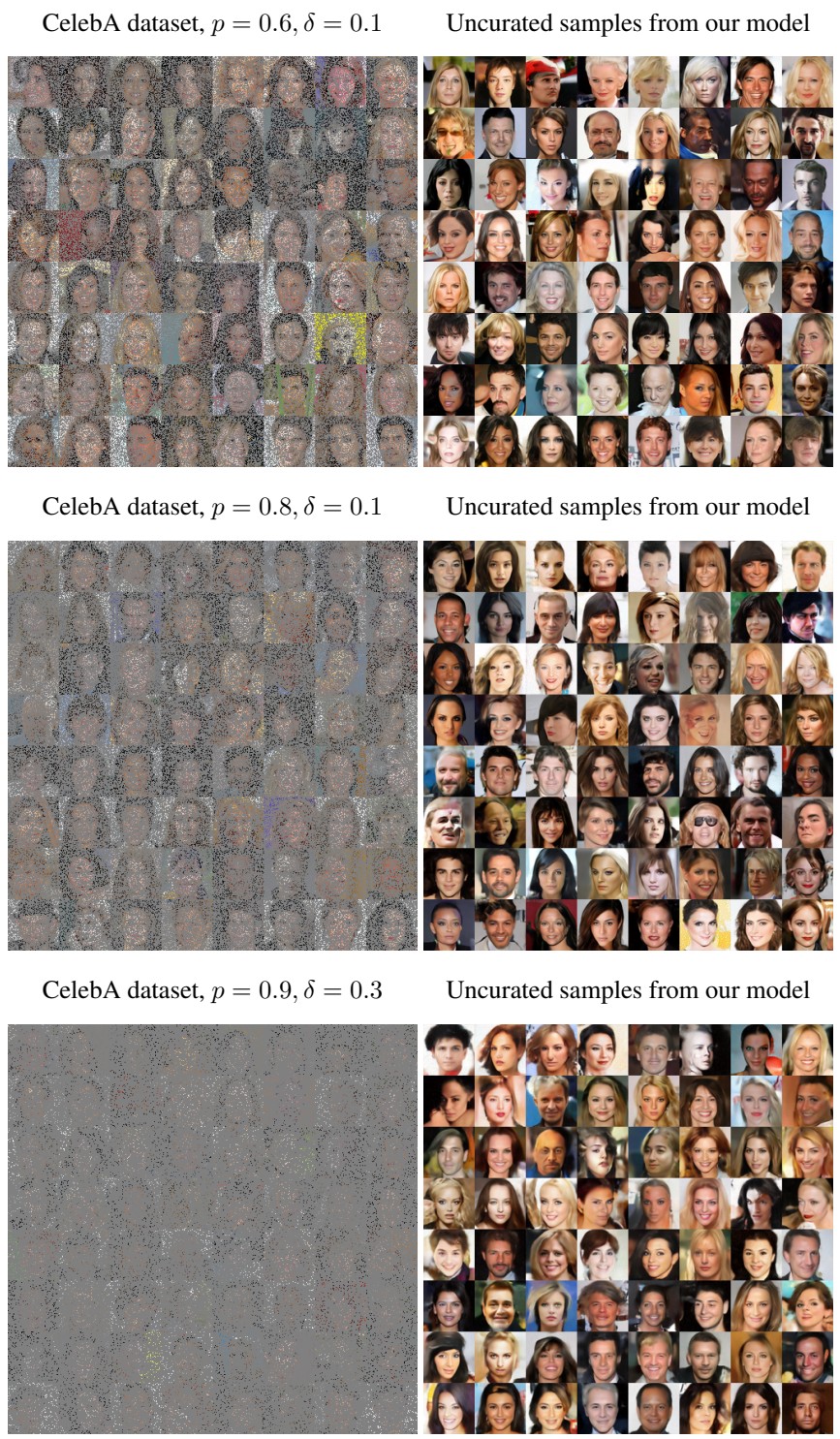

CelebA dataset, $p = 0.6, \delta = 0.1$ — Uncurated samples from our model

CelebA dataset, $p = 0.8, \delta = 0.1$ — Uncurated samples from our model

CelebA dataset, $p = 0.9, \delta = 0.3$ — Uncurated samples from our model

Figure 9: Left column: CelebA-HQ training dataset with random inpainting at different levels of corruption $p$ (the survival probability is $(1 - p) \cdot (1 - \delta)$). Right column: Unconditional generations from our models trained with the corresponding parameters. As shown, the generations become slightly worse as we increase the level of corruption, but we can reasonably well learn the distribution even with 93% pixels missing (on average) from each training image.

AFHQ dataset, $p = 0.4, \delta = 0.1$      Uncurated samples from our model

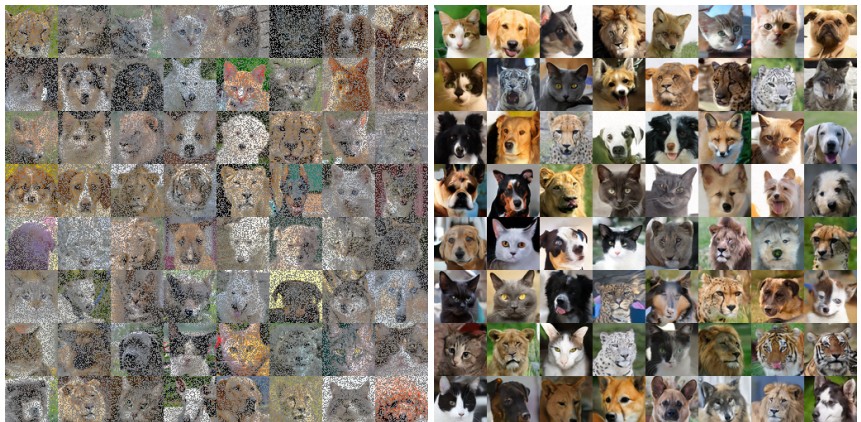

AFHQ dataset, $p = 0.6, \delta = 0.1$      Uncurated samples from our model

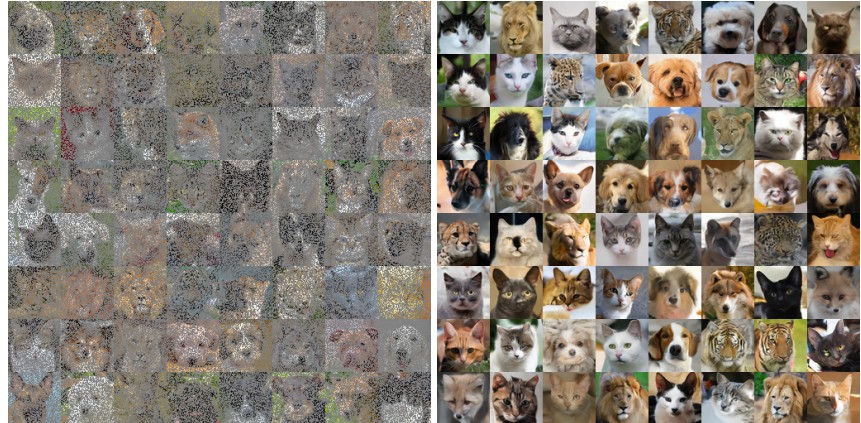

AFHQ dataset, $p = 0.8, \delta = 0.1$      Uncurated samples from our model

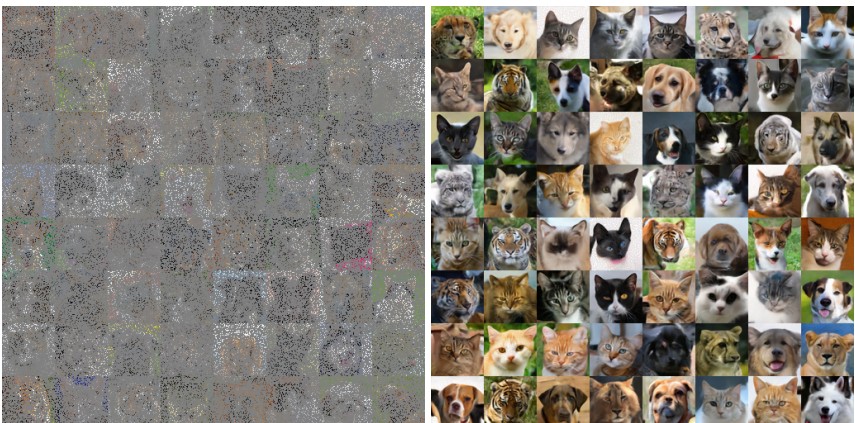

AFHQ dataset, $p = 0.9, \delta = 0.1$        Uncurated samples from our model

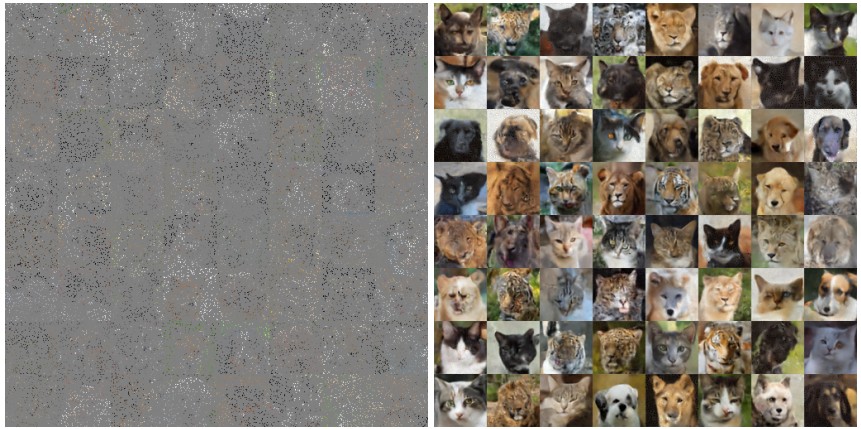

Figure 10: Left column: AFHQ training dataset with random inpainting at different levels of corruption $p$ (the survival probability is $(1 - p) \cdot (1 - \delta)$). Right column: Unconditional generations from our models trained with the corresponding parameters. As shown, the generations become slightly worse as we increase the level of corruption, but we can reasonably well learn the distribution even with $91\%$ pixels missing (on average) from each training image.

CIFAR-10 training dataset, Uncurated samples from our
$p = 0.2, \delta = 0.1$ model

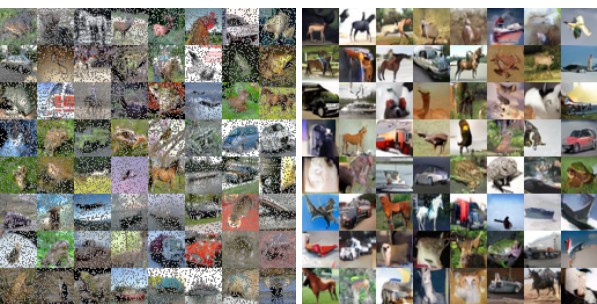

CIFAR-10 training dataset, Uncurated samples from our
$p = 0.4, \delta = 0.1$ model

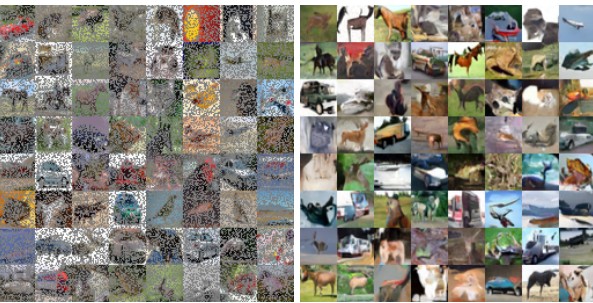

CIFAR-10 training dataset, Uncurated samples from our
$p = 0.6, \delta = 0.1$ model

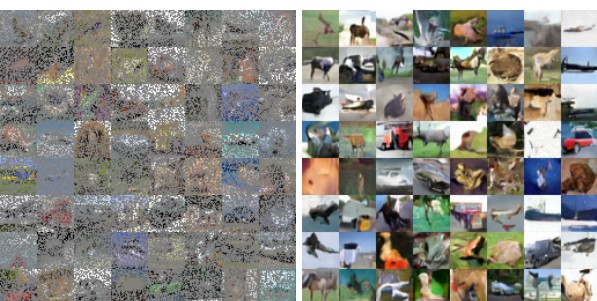

CIFAR-10 training dataset, Uncurated samples from our
$p = 0.8, \delta = 0.1$ model

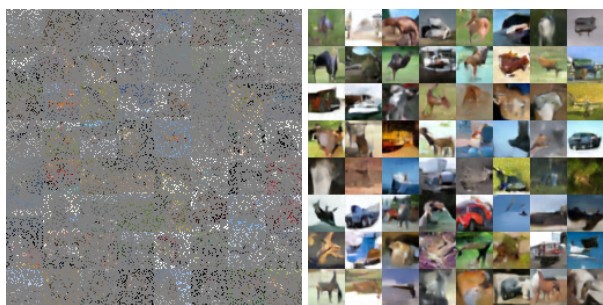

Figure 11: Left column: CIFAR-10 training dataset with random inpainting at different levels of corruption $p$ (the survival probability is $(1 - p) \cdot (1 - \delta)$). Right column: Unconditional generations from our models trained with the corresponding parameters. As shown, the generations become slightly worse as we increase the level of corruption, but we can reasonably well learn the distribution even with high corruption.

