# OpenReview forum: "Ambient Diffusion: Learning Clean Distributions from Corrupted Data"
_NeurIPS.cc/2023/Conference — NeurIPS 2023 poster_

### Official Review · Reviewer_q1F8 · 2023-07-03

**Soundness:** 3 good
**Presentation:** 3 good
**Contribution:** 3 good
**Rating:** 6
**Confidence:** 4

**Summary:**

This paper proposes to train diffusion models that can recover corrupted data without training on clean data. The key idea is, given a corruption matrix $A$ one can further sample a corruption matrix $\tilde{A}$ given $A$ and the model learns to predict all the existing pixels. It is empirically shown that this trick ensures robustness against higher corruption levels and can restore data with better performance than those that's trained on clean data.

**Strengths:**

- The authors propose the first diffusion-based method that can restore data from corruption without training from clean data.

- The method can reuse regular diffusion sampler without much modification.

- It is guaranteed theoretically for the model to recover clean data with some rank assumptions.

- The method alleviates the memorization issues of diffusion models by considering corruption schemes.

**Weaknesses:**

- More exposition is needed on some details. What kind of $\tilde{A}$ is used during inference? Is it necessary to sample using the same $\delta$ during training? Do we fix a $\tilde{A}$ for all sampling steps or resample each step? Does taking the expectation over multiple samples of such $\tilde{A}$ work better?

- The paper lacks implementation details. E.g. $h_\theta(\tilde{A}, \tilde{A}x_t, t)$ explicitly depends on $\tilde{A}$. How is this dependency implemented in practice?

- Line 248 the authors forgot line break.

- The authors only considered random masking corruption. However, there are many other types of linear corruption schemes. Does the method stay robust against other corruption such as Gaussian blur, etc.?

- It would be great to analyze the effect of different sampling scheme for $\tilde{A}$. How does FID change w.r.t $\tilde{A}$ with increasing levels of further corruption? Is there a sweet spot for further corruption?

- Why is there no comparison with AmbientGAN for Table 1, as it is an important baseline?

**Questions:**

My questions and concerns are listed in the section above.

**Limitations:**

- The authors have adequately discussed limitations of the model. Some further discussion on societal impact is encouraged as it is closely related to privacy issues of training data.

---

> ### Author Rebuttal · Authors · 2023-08-09
>
> We thank the Reviewer for their positive and constructive feedback. We are glad that the Reviewer appreciated many aspects of our work, including the novelty of the method, the theoretical analysis, and the implications of reducing memorization.
>
> > More exposition is needed on some details. What kind of $\tilde{A}$ is used during inference? Is it necessary to sample using the same $\delta$ during training? Do we fix a $\tilde{A}$ for all sampling steps or resample each step? Does taking the expectation over multiple samples of such $\tilde{A}$ work better?
>
> We use exactly the same process for generating $\tilde{A}$ during inference. We first sample $A$ from the corruption distribution and $B$ with the same $\delta$ and then set $\tilde{A} = BA$. We cannot use a much different value of $\delta$ during the inference because during the training, the network is trained to take input as the images with missing entries corresponding to $\tilde{A}$.
>
> Regarding fixing or resampling the matrix A, in all of our experiments, we use a fixed $\tilde{A}$ for all sampling steps.
> We tried both approaches and found that the first approach works better. We also tried fixing multiple $\tilde{A}$ for all sampling steps and using an average of $\mathbb E[x_0 | \tilde{A} x_t]$ as an approximation of $\mathbb E[x_0 | x_t]$ and found that this sampler generates blurry images. We will clarify these points in the paper.
>
> > The paper lacks implementation details. E.g.  explicitly depends on $h_{\theta}( \tilde{A}x_t, \tilde{A}, t )$. How is this dependency implemented in practice?
>
> We thank the Reviewer for raising this important clarification question: We concatenate the image with the mask along the channel dimension (e.g. the mask is an extra channel). We will make this clear in the paper.
>
> > The authors only considered random masking corruption. However, there are many other types of linear corruption schemes. Does the method stay robust against other corruption such as Gaussian blur, etc.?
>
> We have only experimented with masking (pixels and blocks), but we do not see any reason why this method wouldn't work for other corruptions, as long as the assumptions needed for our theory are satisfied. We are currently in the process of setting up some non-masking experiments and we will try to include them in the Appendix for the camera-ready version.
>
> > It would be great to analyze the effect of different sampling scheme for $\tilde A$. How does FID change w.r.t with increasing levels of further corruption? Is there a sweet spot?
>
> We thank the Reviewer a lot for raising this question. For a fixed corruption $p$, as we increase $\delta$, we increase the number of missing pixels in the inputs of the network during training. Intuitively, we want to keep $\delta$ as low as possible since we do not want to add additional corruption that is not present in our data. Since the network is trained to predict $\mathbb E[x_0 | \tilde A x_t, \tilde A]$, as $\delta$ goes to $1$, $\tilde A$ becomes the zero matrix and the network only learns the mean of the data distribution. On the other hand, as $\delta \to 0$, the network is not penalized for mistakes in the pixels that are not observed and hence can make arbitrarily wrong predictions in the missing pixels. Empirically, we want to set $\delta$ to the smallest possible value for which the network learns to predict correctly all the missing pixels.
>
> As the Reviewer suggested, there is a sweet spot! To illustrate this, we do a small ablation for the value of $\delta$ for random inpainting in CIFAR-10:
>
> $p$ | $\delta$ | Inception Score
> ------------------------- | ---------  | ------------------
> 0.4 | 0.0 | 6.70 (from Figure 5)
> 0.4 | 0.1 | 7.45 (from Figure 6)
> 0.4 | 0.4 | 6.95 (experiment added for the rebuttal)
>
> As seen, setting $\delta$ to $0$ leads to poor performance: the network does not learn to predict the missing values. Setting a value of $\delta$ to $0.4$ makes sure that the network learns to make accurate predictions everywhere, but now the inputs are more corrupted (only 36% of the pixels survive on average), and hence the predictions are more coarse. Remarkably, the performance of the model with hyperparameters $p=0.4, \delta=0.4$ is only marginally better than the performance for $p=0.6, \delta=0.4$ which has an Inception Score of $6.88$. This is because even though in the former case we have less corruption in our dataset, the effective corruption in the inputs of both models is the same, i.e. only $36%$ of the pixels survive. The optimal value of $\delta$ depends on the dataset and more importantly the resolution – in higher resolution datasets there is more redundancy and we can afford to be more aggressive in the extra corruption. One could try to find the exact value that maximizes performance for a given $p$ (binary search), but due to the high-computational cost of the training, we choose to not ablate this further.
>
> > Why is there no comparison with AmbientGAN for Table 1, as it is an important baseline?
>
> It would be hard, since we would need to train AmbientGANs for these datasets, plus it would be much worse than the baselines we compare against.
> To explain why: AmbientGAN is a framework to train GANs with missing data. Table 1 compares inpainting, i.e. how well we restore a given image with missing pixels, using a pre-trained generator. GANs can be used as priors for image inpainting, but no pre-trained AmbientGANs are available for these large datasets.  Also, diffusions perform better compared to GANs for inpainting and other inverse problems. Therefore, beating state-of-the-art pre-trained diffusion models (trained on clean data!) used for inpainting, is a harder task than beating AmbientGAN-guided inpaining. Table 1 shows that we can match the performance or even outperform these stronger baselines even without having access to clean data. We hope this clarifies things and we plan to include this discussion to make it more clear in the paper.

---

> > ### Author Response · Authors · 2023-08-21
> >
> > As the discussion period approaches its end, we would like to kindly ask if the Reviewer had a chance to read our rebuttal. We thank again the Reviewer for their time and their constructive feedback in their initial review and we hope that our rebuttal addressed the Reviewer's questions.

---

> > > ### Comment · Reviewer_q1F8 · 2023-08-21
> > > **Thank you for the rebuttal**
> > >
> > > The authors have resolved my concerns. I have raised my score.

---

### Official Review · Reviewer_Tq9h · 2023-07-05

**Soundness:** 3 good
**Presentation:** 3 good
**Contribution:** 3 good
**Rating:** 7
**Confidence:** 2

**Summary:**

In summary, the authors propose a diffusion-based framework that can learn unknown distributions from highly-corrupted samples, allowing the training of generative models without relying on clean training data. Their approach introduces additional measurement distortion and successfully predicts original corrupted images. The method is applicable to various corruption processes and achieves promising results on benchmark datasets.

**Strengths:**

- The problem formulation by itself is interesting, and the proposed method is novel.
- The paper proposes a new and interesting domain of learning the image data distribution w/o access to the ground truth data samples.
- The paper is well written and easy to follow.
- The proposed training and sampling procedure is scalable and easy to incorporate into the current diffusion model framework.
- The authors attach the theory for the effectiveness of the proposed method.
- The proposed method only needs 1 NFE to produce comparable results.
- The method paves a way to alleviate the memorize issue of diffusion model.

**Weaknesses:**

- From my perspective, I do not see significant weaknesses in this paper.
- One potential issue is the significant drop of FID when the model is trained with images corrupted with a large ratio of pixels. However, I think this should not be criticized.

**Questions:**

- Can you provide the performance of the proposed method in Table 1 with more timesteps of sampling? Does the performance improve? If no, then can you please explain why?


**Limitations:**

The author lists the limitation of the proposed method in the paper which is reasonable. I appreciate the authors' adequate limitation summary and it sheds light on future improvements.

---

> ### Author Rebuttal · Authors · 2023-08-09
>
> We are very glad that the Reviewer appreciated the importance of the problem, the novelty, the presentation, and the theoretical and practical implications of our work!
>
> > Can you provide the performance of the proposed method in Table 1 with more timesteps of sampling? Does the performance improve? If no, then can you please explain why?
>
> We thank the Reviewer for the great question. We achieve optimal reconstruction in one step because we learn the conditional expectation (as predicted by our theory) which is the best reconstruction under the $l_2$  loss.
>
> We thank again the Reviewer for their positive feedback and we would be happy to take further questions, if any.

---

> > ### Comment · Reviewer_Tq9h · 2023-08-16
> > **Response to Authors**
> >
> > Thank you for your answer. I will keep the current score.

---

### Official Review · Reviewer_qLWX · 2023-07-05

**Soundness:** 2 fair
**Presentation:** 2 fair
**Contribution:** 3 good
**Rating:** 6
**Confidence:** 4

**Summary:**

This paper describes a method to learn a denoising diffusion model only with corrupted data. This is an important problem in many areas of  applied science where there is no access to ground truth. Another important potential benefit of this method is to overcome memorization of the training images. The main idea in this method is described as introducing additional corruption and training on this doubly corrupt data. The authors describe a derivation to estimate conditional expectation of the uncorrupted image only using the corrupted data. They show (Table 1) that their method outperforms other methods in solving random inpainting problem. Also, the method is used to fine tune Deepfloyd IF on smaller samples. This result is used to show that the method overcomes memorization.

**Strengths:**

This work is motivated in two ways: learning the score of distribution of clean data with access to corrupted data, and avoiding memorization. These are both very important practical and theoretical topics which this work tries to address. The trick which is used to train the models (adding more corruption) is clever and the network achieves to learn to inpaint and denoise images (figure 7). As mentioned under weaknesses, I think the first goal is not fully achieved and the theoretical results seem incorrect and practical results seem to be constrained to specific types of corruption. However, the memorization results seem very impressive, although the memorization analysis is carries only for the fine tuned model.

**Weaknesses:**

There are a number of fundamental technical and conceptual flaws which need to be considered and fixed:

1) The training setup was not clear in the text, but this is what I gathered: the target image during training is a corrupted image $Ax_0$, and the input is a noisy and more corrupted image $BAx_0 + \sigma \eta $. All throughout the corruptions are assumed to be random or block missing pixels. The network learns to remove the noise and inpaint pixels that are removed by B. At the test time, the network removes noise and also inpaint *all* pixels since it doesn't know which pixels are dropped due to A and which pixels are dropped due to B.
If this is the training setup, please clarify in the text. If not, please describe what was the training setup.

2) Equation (3), the objective after additional corruption, is the distance between the doubly corrupted image and the $\textbf{clean}, x_0,$ image. The entire premise of the work is that clean image is not available so why is it assumed to be available during training? The objective should be the distance from $Ax_0$ instead. It needs to be clarified whether this is a typo or the authors actually used clean data, $x_0$, during training.

2) If that is a typo and the target image during training is $Ax_0$ then after training is completed the network works as a denoiser plus inpainter. As a result, the Tweedie equation is not a good description of the output of this network anymore. That is, the output is not $E[x_0|x_t]$, where $x_t = x_0 + \sigma_t \eta$. So, this output cannot be used directly as an estimate of score in the diffusion model .

3) To remedy the above mentioned problem, the authors propose eq 3.3 in which they claim to approximate the score, $E[x_0|x_t],$ with $E[x_0|\tilde{A}x_t, \tilde{A}]$. It is not at all clear what is the justification behind this approximation. Again, this expectation is a very different entity from the actual score, and it is not clear why direct use of it makes sense to estimate the score. This expectation is the solution for inverse problem given the forward measurement $A$, as apposed to the solution for mere denoising problem (i.e. the score).

4) Additionally, two more terms are added to the update line in eq 3.4. The justification for this is described from line 166 to 175. It starts with $\gamma_t$ going to zero when $t$ approaches zero. It is not clear as to why $\gamma_t$ goes to zero when $t$ goes to zero. Please clarify this assumption. The reasoning that follows this assumption is also not clear and sounds ad hoc. Did you add these terms because the update line of eq 3.3 did not work in practice? What is the intuition or theory behind this choice? Please clarify both eq 3.3. (why did you estimate one expectation with another?) and eq 3.4 (why did you add two terms and why do they make sense?).

5) The theory section needs at least a re-write because it is not clear what is the goal of this section. The section starts with the goal of proving that the optimal estimate of clean image given the corrupted image is equal to $E[x_0| Ax_t = y, A]$. It is not clear why the authors need to prove this, since this is a basic fact from Bayesian machine learning: the optimal estimation of corrupted data is the conditional mean of the predictive distribution (refer to textbooks like Bishop). The problem is that this expectation is not equal to the score, so theoretically it can't be used to estimate the score. Of course what is learned by this network is approximately the optimal reconstruction of $A(x+\sigma \eta)$, but as long as it is not the score, it is not theoretically valid to use it iteratively in a diffusion model. In a nutshell, the learned network is a denoiser/inpainter not a score estimator (i.e. pure denoiser).

6) On top of the above mentioned issues, this network learns to reconstruct + denoise images only if $E_{A|\tilde{A}}[A^TA]$ is full rank. This strong assumption requires a high level of randomness in the corruption which is not very common in many real-world applications. For example, if the ground truth images miss some information systematically (let's say they are all blurred) the network will not be able to reconstruct.

**Questions:**

In addition to theoretical questions and comments I made under weaknesses:

Why do you think you achieve optimal sampling performance in one iteration?
How does the training time compares to uncorrupted models? Do you need to train on more samples?
________________________________________________________
Note: I am willing to raise the score if the questions asked under Weaknesses and here are addressed.

**Limitations:**

The main limitation of the work (aside from some jumps  and ambiguity in the theoretical results) is the type of corruption this method can work with. The corruption must be linear and in addition $E_{A|\tilde{A}}[A^TA]$ must be full rank. This rules out many real world applications.

---

> ### Author Rebuttal · Authors · 2023-08-09
>
> We thank the Reviewer for their feedback!
>
> > Equation (3), [...] is the distance between the doubly corrupted image and the clean image. The objective should be the distance from $Ax_0$.  It needs to be clarified whether this is a typo or the authors actually used clean data.
>
> The Reviewer has misread Equation 3. *There is a parenthesis outside*, A multiplies both the output of the network and the clean image. Hence, the only thing needed to train is the corrupted image Ax and not the clean image itself. **We do not use clean data during training.**
>
> > The theory section needs at least a re-write because it is not clear what is the goal of this section. [...] It is not clear why the authors need to prove this, since this is a basic fact from Bayesian machine learning [...].
>
> The Reviewer may have a misunderstanding regarding our theoretical results, potentially because of misreading Equation 3.
> Proving that the minimizer of  $\mathbb E_{x_0,Ax_t}[ || f(Ax_t) - x_0||^2]$  is the conditional expectation is a textbook argument. However, this is *not* what we prove. We prove something much stronger which is that there is an objective function (Eq. 3.2) that doesn’t need access to the clean images $x_0$ and still has as a minimizer the conditional expectation (Theorem 4.1). Our proof does build on the standard techniques but it has the key benefit of making clear what condition we need on the distribution of $A, \tilde{A}$ for the theorem to hold. Thus we argue that our result is novel, and we think that the Reviewer may not have appreciated it fully. We kindly ask the Reviewer to reconsider their evaluation, keeping in mind that our loss does not require clean images and that the proof technique has some differences compared to the standard setting. We would be happy to take further feedback if there are additional concerns.
>
>
> > If this is the training setup, please clarify in the text. If not, please describe what was the training setup.
>
> The Reviewer’s understanding of the training setup is correct. We will clarify this further in the main text.
>
> > [...] This expectation is a very different entity from the actual score, and it is not clear why direct use of it makes sense to estimate the score.
>
> It is true that we are making an approximation there, as we acknowledge in the paper. We approximate $\mathbb E[x_0 | x_t]$ with $\mathbb E[x0 | Axt, A]$. Intuitively, if A is not dropping a lot of information, these two conditional expectations are close. This approximation becomes worse for high corruption. This is a limitation of this work. In Lemma A.3. we prove that access to $\mathbb E[x_0 | Ax_t, A]$ is enough to reconstruct the true distribution whenever it is possible to reconstruct it from measurements. However, for the time being, we do not have an efficient algorithm to do so and thus we resort to this approximation.
>
> > two more terms are added to the update line in eq 3.4. [...] It is not clear as to why $\gamma_t$ goes to zero when t goes to zero. [...] Did you add these terms because the update line of eq 3.3 did not work in practice? Please clarify both eq 3.3. and eq 3.4.
>
> We think that the Reviewer has a misconception here. The $\gamma_t$ appears already from Equation 3.3 and it is not something that we added, it is the term that appears in all diffusion models when you discretize the reverse SDE/ODE. It always goes to zero and it stems from the fact that the magnitude of the noise is an increasing function of time and at the limit $t\ to 0$ we do not have any noise. The only thing difference to the classic ODE discretization is that we have $\mathbb E[x_0 | Ax_t, A]$ instead of $\mathbb E[x_0 | x_t]$.
>
> The Reconstruction Guidance sampler adds an additional gradient update that refines all pixels throughout the denoising process. The Fixed Mask sampler uses a fixed mask $A$ and thus the masked pixels are not getting refined at every iteration. Hence, for high corruption, these pixels get more “averaged” values. The reconstruction guidance sampler mitigates this. We ablate this Sampler in Table 5 in the Appendix and we see modest improvements that were not worth the extra computational cost hence we did not use this sampler in any experiment in the main paper. We hope this clarifies things.
>
> > This network learns to reconstruct + denoise images only if $E_{A|\tilde A}[A^TA]$ is full-rank. This strong assumption requires a high level of randomness in the corruption which is not very common in many real-world applications.
>
> Without this assumption, it is *impossible* to reconstruct the distribution from corrupted samples without making assumptions on the distribution to be reconstructed. This is not a limitation of our method; any method that tries to recover the distribution from corrupted samples would fail. The reason is that it is impossible to distinguish between two clean distributions that become identical after the pushfoward function. E.g., in the blurring example, depending on what is the blurring kernel, there could be many distributions that lead to the same blurred measurements.
>
> > Why do you think you achieve optimal sampling performance in one iteration?
>
> We achieve optimal reconstruction in one step because we learn the conditional expectation which is the best reconstruction under L2 loss.
>
> > How does the training time compares to uncorrupted models? Do you need to train on more samples?
>
> We thank the Reviewer for this great question! The training time and the dataset size are the same as for the training of the uncorrupted models. All models are trained for 200K steps (following the EDM paper) and we use the full dataset. For the finetuning experiments, we show that we can even finetune with 300 images or less. We refer to the Appendix, Section C, for the full training details.
>
> We hope that the Reviewer's comments are addressed and the Reviewer will consider increasing their score as they noted in their review.

---

> > ### Comment · Reviewer_qLWX · 2023-08-17
> >
> > Thank you for responding to my comments and questions.
> >
> > The comment about eq 3.2 was indeed due to my misreading of it. Thanks for clarification. In my initial review, as I had mentioned, I had assumed the apparent missing A was a typo and not a conceptual error. So the rest of my initial review was based on the assumption that you had the correct form of the eq 3.2 (as you did).
> >
> > Regarding the theory section, I am convinced now that Theorem 4.1 is valuable beyond what I was evaluated initially. However, it relies on the strong assumption on full rank $E[A^TA]$, so a discussion on this assumption should be included in the text. This assumption excludes many real work corruptions. It should be mentioned earlier in the text (in the intro or even abstract). At its current state, the text makes the impression that this method magically learns the score from any corrupted data, until section 4. It can be mentioned earlier that this method works as long as the corruption is not systematic. For more discussion, refer to literature on Stein Risk Estimator (SURE) in which the minimizer is estimated without access to clean data for the case of Gaussian corruption. Your proof can be thought of as a generalization of this old idea.
> >
> > It is very interesting that the corruption does not require training on more data points. I think it is worth to include this in the main text.
> >
> > A minor note: do not assume that all readers are familiar with the notations you use. Explain in the text what $\gamma$ is. The paper should be self-sufficient in terms of notations and definitions.
> >
> > Regarding the final update line, eq 3.3 and eq 3.4, I still find the discussion unsatisfactory. This is at the core of the method, and I am not convinced why this crude approximation (in 3.3) is acceptable. Of course if the corruption is small, the approximation would be not too far, but that is not what this paper claims to do. Additionally the term added in 3.4 is not well motivated. Again, if there is a good motivation from the paper you're borrowing this from, it should be described and explained in the text. Why is that term a good addition to your update line? At it's current state, there is no clear connection between the theory section and this added term.
> >
> > I raise my score from 4 to 5 since the authors addressed some of my questions. There is still room for improvement regarding the above point (point 3 and 4 in my initial review).

---

> > > ### Author Response · Authors · 2023-08-19
> > > **Additional Discussion**
> > >
> > > We thank the Reviewer for engaging in the discussion and for raising their score. Their valuable time is deeply appreciated!
> > >
> > > We are glad that the issue that arose from the misreading has now been resolved, that the Reviewer appreciated our Theorem 4.1 and that the Reviewer found satisfying that we didn’t use more data points for the corrupted models. We will definitely include this in the main text, as recommended by the Reviewer.
> > >
> > > We will also make it more clear from the Introduction that our method does not work for arbitrary corruptions. In fact, there is no method that learns the true distribution for arbitrary corruptions. The assumption we make on the corruption process is necessary to learn from inpainted data – if we never observe certain pixel locations there is no way to identify the distribution in these locations unless we make assumptions on the distribution itself. As we pointed out in our rebuttal, previous methods such as AmbientGAN, make such assumptions (e.g. from AmbientGAN: “A critical assumption for our framework and theory to work is that the measurement process is known and satisfies certain technical conditions”).
> > >
> > > These assumptions are not very restrictive when we control the corruption process (e.g. to reduce memorization) but preclude (theoretically) applying our method for non-systematic corruption processes, as the Reviewer noted correctly. We will clarify this very early in the text, as recommended by the Reviewer. We also want to point out that our method could potentially achieve reasonable performance in practice for corruption processes that violate such assumptions, but in this case it does not come with any theoretical guarantees.
> > >
> > > We will also further highlight how our proof adds to the literature of Stein Risk Estimation (SURE) and generalizes this to the case of non Gaussian corruption. We thank the Reviewer for recommending this. Finally, we will add the explanation from our rebuttal to the main text regarding what $\gamma_t$ is to avoid potential confusions.
> > >
> > >
> > > Regarding the remaining concern of the Reviewer about the sampling, we want to start by acknowledging again that we are making an approximation there. We clearly state this in the Limitations Section of our work (“Further, in this work we only experimented with very simple approximation algorithms to estimate $\mathbb E[x_0|x_t]$ using our trained models”) and in the Method Section (“we approximate $\mathbb E[x_0|x_t]$ given the predictions of $\mathbb E[x_0|Ax_t, A]$”). Surprisingly, this simple approximation outperforms previous baselines and works reasonably well even for high corruptions (potentially because there a lot of redundancies in natural images). We are planning to explore in future work how to make this step exact and we might get a significant performance boost (especially in the high corruption regime) by fixing this.
> > >
> > > Regarding the term in the Equation 3.4, this is inspired by the video diffusion models literature. For video generation with diffusion models, the models are trained to denoise the current frame but are regularized so that denoised version is not too different compared to the prediction for neighboring frames. This heuristic is needed for smooth transitions in the video generation process. Similarly, in our setup, we only have access to models that predict given limited context and we want to combine the predictions such that there are no artifacts or inconsistencies. The fixed mask sampler ensures there are no artifacts by always masking the same pixels. In Eq. 3.4, we added this extra term to (heuristically) remedy that the values in the masked pixels in the fixed mask sampler only depended on the final prediction – we update all pixels at every prediction using many different masks and ensure consistency of the different predictions through this added term. Again, this is a heuristic step and future work might significantly improve performance by making this exact, thus we understand if the Reviewer finds this approximation unsatisfactory. We did not end up using this sampler for the experiments in the main paper due to the extra computation needed in return for marginal benefits, e.g. see Appendix E.3 and Table 5. Since the Reviewer did not appreciate this attempt to heuristically improve the sampling process, we can move this entirely to the Appendix (together with the discussion above) and make space for the rest of the requested changes that are more critical to the paper.
> > >
> > >
> > >
> > > We hope this discussion is useful and we want to thank again the Reviewer for helping us improve our paper!

---

> > > > ### Comment · Reviewer_qLWX · 2023-08-21
> > > >
> > > > Thanks for clarifying the point about the added term in eq 3.4. I believe it will significantly improve the clarity of the paper if, as you suggested, this is moved to the appendix. Or at least a similar description of the heuristic is offered in the text. In its current state, it seems very arbitrary.
> > > >
> > > > Although the approximation of $E[x_0|x_t]$ by $E[x_0|Ax_t,A]$ is unjustified, I think this paper overall contains significant contributions to the field. Thus I raise my score to 6.

---

### Official Review · Reviewer_GQa2 · 2023-07-07

**Soundness:** 3 good
**Presentation:** 3 good
**Contribution:** 3 good
**Rating:** 6
**Confidence:** 4

**Summary:**

The paper focuses on learning clean distributions from corrupted data. In the training diffusion model, the training dataset contains only highly-corrupted examples. They propose a training algorithm of restoration model by introducing additional measurement distortion.They also provide sampling methods and theoretical analysis. Experimental results show that their superior performance.

**Strengths:**

* The problem of handling corrupted datasets is significant even in generative model learning. This paper could be seen as one that explores this directions.
* The paper is well-structured and easy to follow.
* The fact that a model which is trained by corrupted data does not memorize the training dataset, as supported by Figure 1 and Figure 4, is very important. This might be gives significant implications for various applications.

**Weaknesses:**

* Sampling Process
  * It would be helpful to provide an algorithm or a detailed explanation of the sampling process from scratch.
  * Equations 3.3 and 3.4 describe how the sample at time $t$ is generated from sample at $t-\Delta t$. It is necessary to explain how the evaluation of single networks allows the generation of images in the experiments.

* Related work
  * It would be beneficial to include a discussion of previous research on dealing with incomplete datasets in generative models, such as [1, 2, 3, 4].

[1] Li, S. C. X., Jiang, B., & Marlin, B. (2018, September). MisGAN: Learning from Incomplete Data with Generative Adversarial Networks. In International Conference on Learning Representations.

[2] Mattei, P. A., & Frellsen, J. (2019, May). MIWAE: Deep generative modelling and imputation of incomplete data sets. In International Conference on Machine Learning (pp. 4413-4423). PMLR.

[3] Ipsen, N. B., Mattei, P. A., & Frellsen, J. (2020, September). not-MIWAE: Deep Generative Modelling with Missing not at Random Data. In International Conference on Learning Representations.

[4] Richardson, T. W., Wu, W., Lin, L., Xu, B., & Bernal, E. A. (2020). Mcflow: Monte carlo flow models for data imputation. In Proceedings of the IEEE/CVF Conference on Computer Vision and Pattern Recognition (pp. 14205-14214).

* Experiments
  * It would be beneficial to provide the FID results for Figure 5 to demonstrate the differences.
  * In addition to Figure 5, it would be valuable to compare the proposed model with existing generative models such as AmbientGAN in various experiments.

* Presentation
  * It would be helpful to have better spacing between subfigures in Figure 2 to improve the clarity of the captions.
  * Proper citations are needed in the background section.
  * Figure 3 would be better placed within a paragraph rather than between paragraphs.

**Questions:**

Please see Weaknesses part.

**Limitations:**

They provided in the last paragraph in the main paper.

---

> ### Author Rebuttal · Authors · 2023-08-09
>
> We thank the Reviewer for the constructive feedback! We are glad that the Reviewer appreciated the novelty, the presentation and the implications our work could have in various applications related to memorization.
>
> > It would be helpful to provide an algorithm or a detailed explanation of the sampling process from scratch.
>
> We agree with the Reviewer. Please see the detailed sampling algorithm in the one page PDF accompanying this rebuttal. We will include this algorithm in the next revision of our work.
>
> > it is necessary to explain how the evaluation of single networks allows the generation of images in the experiments.
>
> This becomes clear with the sampling algorithm we include in our rebuttal. This works by a process of iterative restoration-degradation, similar to how all diffusion models operate.
>
> > It would be beneficial to include a discussion of previous research on dealing with incomplete datasets in generative models, such as [1, 2, 3, 4].
>
> We thank the Reviewer for bringing this relevant work to our attention! We will definitely add these references in the camera-ready version of our work. All these works propose ways to learn other classes of generative models (such as GANs, Normalizing Flows and VAEs) from missing data. The MisGAN work generalizes AmbientGAN in the case where the measurement operator is unknown. Specifically, the authors propose an additional generator that learns to model the corruption mechanism with adversarial training. MCFlow is a framework, based on a variant of the EM algorithm, that can be used to train normalizing flow models from missing data. Finally, MIWAE and Not-MIWAE are frameworks to learn deep latent models (e.g. VAEs) from missing data when the corruption process is known or unknown respectively. Our work provides a diffusion-based framework for the missing data problem and thus expands this interesting prior work.
>
>
> > It would be beneficial to provide the FID results for Figure 5 to demonstrate the differences.
>
> Figure 5 shows the Inception results for CIFAR-10. The FID results for the same dataset are shown in Figure 6. Unfortunately, the authors of AmbientGAN do not report FID scores.
>
> > In addition to Figure 5, it would be valuable to compare the proposed model with existing generative models such as AmbientGAN in various experiments.
>
>
> We thank the Reviewer for the suggestion. AmbientGAN only provides quantitative results in MNIST and CIFAR-10 (Figures 7 and 8 in the AmbientGAN paper). We have the comparison with CIFAR-10 (Figure 5) in the paper. We did not experiment on the MNIST dataset since it is a rather toy problem for image generation.
> The Reviewer brought to our attention the follow-up work to the AmbientGAN paper, MisGAN. The authors of MisGAN report FID scores for different erasure probabilities for CIFAR-10 and CelebA. We will include these comparisons in the next version of our paper. A short comparison is provided below:
>
>
> **CelebA**:
> Corruption Probability | Method | FID |
> --------------------------| --------- | ---- |
> 0.6                               |    MisGAN      |  37.42
> 0.6                                    | Ambient Diffusion | **6.08**
> 0.8                              |    MisGAN      |  100.0
> 0.8                                    | Ambient Diffusion | **11.19**
> 0.9                               |    MisGAN      |  141.11
> 0.9                                   | Ambient Diffusion | **25.53**
>
>
> **CIFAR-10**:
> Corruption Probability | Method | FID |
> --------------------------| --------- | ---- |
> 0.4                               |    MisGAN      |  18.95
> 0.4                                    | Ambient Diffusion | **18.85**
> 0.6                              |    MisGAN      | 49.30
> 0.6                                    | Ambient Diffusion | **28.88**
> 0.8                               |    MisGAN      |  111.50
> 0.8                                   | Ambient Diffusion | **46.27**
>
> We will include this comparison in the camera-ready version of our work. It can also be found in the one page PDF accompanying this rebuttal. We emphasize that MIsGAN is solving a harder problem than we are, since for MisGAN the corruption operator is not known and needs to be inferred-- we will explain this in the paper.
>
> We finally want to thank the reviewer for the comments in the presentation of our work. We will make sure to add the additional citations, fix the spacing and improve the placement of Figure 3, as suggested.

---

> > ### Comment · Reviewer_GQa2 · 2023-08-16
> >
> > Thank you for your response and further comparison. My concerns are mostly addressed, so I'm raising my rating from 5 to 6.

---

### Author Rebuttal · Authors · 2023-08-09

We thank the Reviewers for their constructive feedback! We are very glad that our work was well-received and that the novelty, the experimental and the theoretical contributions were generally appreciated by the Reviewers.

We include separate replies to each one of the Reviewers.

We also attach a one-page PDF that contains additional experiments and a formal statement of our sampling algorithm, as requested by some of the Reviewers.

We remain available to answer additional questions if any!

---

### Decision · Program_Chairs · 2023-09-21

**Decision:**

Accept (poster)

**Comment:**

The authors propose a diffusion-based framework for learning unknown distributions from heavily corrupted samples. This addresses scenarios in scientific applications where access to clean samples is infeasible or costly. A key idea involves introducing "additional measurement distortion" during diffusion and training the model to predict the original corrupted image from the further corrupted version. The authors also validate their approach on standard benchmarks (CelebA, CIFAR-10, AFHQ), showcasing the ability to learn distributions even when training samples have significant pixel gaps.

All reviewers thought this is a nicely written paper and interesting despite some limitations. I concur and recommend acceptance.